# MMD Two-sample Testing in the Presence of Arbitrarily Missing Data

**Yijin Zeng** *yijin.zeng20@imperial.ac.uk*
*Imperial College London*

**Niall Adams** *n.adams@imperial.ac.uk*
*Imperial College London*

**Dean Bodenham** *dean.bodenham@imperial.ac.uk*
*Imperial College London*

**Reviewed on OpenReview:** *https://openreview.net/forum?id=GfcDel1ICb*

## Abstract

In many real-world applications, it is common that a proportion of the data may be missing or only partially observed. We develop a novel two-sample testing method based on the Maximum Mean Discrepancy (MMD) which accounts for missing data in both samples, without making assumptions about the missingness mechanism. Our approach is based on deriving the mathematically precise bounds of the MMD test statistic after accounting for all possible missing values. To the best of our knowledge, it is the only two-sample testing method that is guaranteed to control the Type I error for both univariate and multivariate data where data may be arbitrarily missing. Simulation results show that the method has good statistical power, typically for cases where 5% to 10% of the data are missing. We highlight the value of this approach when the data are missing not at random, a context in which either ignoring the missing values or using common imputation methods may not control the Type I error.

## 1 Introduction

Two-sample hypothesis testing is a fundamental statistical method used to determine if two samples of data are different. Numerous two-sample testing methods are available, including Student's $t$-test (Student, 1908), the Wilcoxon-Mann-Whitney $U$ test (Mann & Whitney, 1947), the Kolmogorov-Smirnov test (Kolmogorov, 1933), the Energy Distance (Rizzo & Székely, 2016), and the Maximum Mean Discrepancy (MMD) test (Gretton et al., 2012a). These methods have proven useful in various fields such as medicine (Tsiatis et al., 2008; O'Brien & Fleming, 1979), finance (Lukić & Milošević, 2024; Dave & Korkie, 1981), psychology (Pfister & Janczyk, 2013) and machine learning (Gretton et al., 2012a; 2009; Schrab et al., 2023). Nearly all two-sample testing methods are designed solely for data that are fully observed, with a few exceptions discussed in the next section. However, in many real-world datasets, a subset of univariate values may be missing or multivariate values may only be partially observed. For example, in clinical trials it is common that patients drop out during the trial (Bakris et al., 2015), resulting in missing data for those participants.

When data are missing, common practices are either to ignore all missing values or impute these values using some imputation schemes. Following this step, the data are treated as complete, allowing any standard two-sample testing method to be used. However, except in special cases, such as when the missing data are missing completely at random (Rubin, 1976), these practices are often invalid as they risk increasing the probability of a Type I error occurring. Under certain assumptions, such as when the data are missing completely at random, missing at random, or when the data are missing not at random but the missingness mechanisms (the

samples can be subject to different missingness mechanisms) can be explicitly specified (Schafer & Graham, 2002), then certain sophisticated missing data methods exist, such as the expectation-maximization algorithm and the multiple imputation method (Schafer, 1999). However, without these assumptions, relying on these methods is fraught with risk.

Recently a two-sample testing method for data with univariate missing values was proposed which does not rely on a specific imputation procedure (Zeng et al., 2024). This method, based the Wilcoxon-Mann-Whitney test, rejects the null hypothesis after accounting for all possible values of the missing data by deriving bounds for the $U$ statistic. This approach avoids ignoring or imputing the missing data and is shown to control the Type I error without making assumptions about the missingness mechanisms, while also having good statistical power when the proportion of missing data is around 10%. However, this approach is restricted to univariate samples and can only detect location shifts since it is based on the Wilcoxon-Mann-Whitney test.

In this paper, we propose a novel two-sample testing method based on the MMD test in the presence of missing data that makes no assumption about the missingness mechanisms, following the approach in Zeng et al. (2024). However, our approach can be used for both univariate and multivariate samples, and can detect any distributional change since it is based on the MMD test.

Our approach, named **MMD-Miss**, is based on deriving the bounds of the unbiased MMD test statistic (Gretton et al., 2012a) in the presence of missing data. To do so, we use the Laplacian kernel, a popular choice of characteristic kernel that enables MMD to detect any distributional shift (Sriperumbudur et al., 2011). To compute the $p$-value for the test, we either use Monte Carlo sampling from the permutation distribution of the observed data or, in the case of high-dimensional multivariate data, we use a normality approximation recently derived in (Gao & Shao, 2023). We prove that our approach controls the Type I error, regardless of the values of the missing data. Numerical simulations are presented, showing that our method has good testing power when the proportion of missing data is around 5% to 10%, while controlling the Type I error.

**Motivation and contribution.** Our work is motivated to ensure valid two-sample testing in the presence of missing data without relying on missingness assumptions. In the context of performing two-sample testing with missing data, a common misunderstanding is that when the proportion of missing data is small, e.g. 5% as suggested by Schafer (1999), and Heymans & Twisk (2022), the testing result will not be skewed by missing data after either ignoring or imputing these values. While these practices may be justified under certain conditions, such as when the data are missing completely at random (Rubin, 1976), we provide experiments in Figure 1 showing that when the data are missing not at random, these practices can lead to a Type I error asymptotically equals to 1 even with a small proportion (5%) of missing data. Our contribution is a method based on the MMD two-sample test that is suitable for univariate and multivariate data with missing values. In contrast to other methods, this approach can detect any change in distribution while being guaranteed to control the Type I error, and without making any assumptions about the missing data.

## 2 Background

This section provides the necessary background for our work. We begin by introducing the two-sample testing problem and then review the different types of missingness mechanisms. Next, we briefly review the MMD two-sample test, and then discuss two standard approaches for implementing the MMD test: the permutation test and the normal approximation. Finally, we highlight related work with a focus on existing approaches to two-sample testing in the presence of missing data as well as extensions of the MMD test.

**Two-Sample Testing.** A two-sample testing method is used to determine whether two groups of data are statistically significantly different. Consider two samples of observations $X = \{x_1, \ldots, x_{n_1}\}$ and $Y = \{y_1, \ldots, y_{n_2}\}$, where each observation is in $\mathbb{R}^d$, and suppose that $X$, and $Y$ are independent random samples from distributions $p$, and $q$, respectively. We define the null hypothesis $H_0$ as $p = q$ and the alternative hypothesis $H_1$ as $p \neq q$. The two-sample testing method first computes a statistic based on the data, and then a $p$-value based on the statistic. After comparing the $p$-value to a pre-specified significance threshold $\alpha$, the null hypothesis $H_0$ is either rejected or fails to be rejected.

A Type I error occurs when the null hypothesis $H_0$ is true but the test incorrectly rejects it. Conversely, a Type II error occurs when $H_0$ is false but the test fails to reject it. A two-sample testing method is usually derived so that, under $H_0$, the Type I error is not larger than the pre-specified significance threshold $\alpha \in (0, 1)$. In this case, we say the test controls the Type I error. For a given $\alpha$, a two-sample testing method is preferred if it has a lower probability of making a Type II error, denoted by $\beta$. The power is defined as $1 - \beta$, and a two-sample testing method is more desirable if it demonstrates higher power. Two-sample testing methods are usually assessed on their power, given that the Type I error is controlled.

**Missingness Mechanisms.** Missingness mechanisms are typically classified as missing completely at random (MCAR), missing at random (MAR), and missing not at random (MNAR) following Rubin (1976). Let $Z_{\text{com}} = (z_1, \ldots, z_n)^T$ denote the complete data, where each $z_i$ is a $\mathbb{R}^d$ real value. The complete data $Z_{\text{com}}$ can be split into observed parts $Z_{\text{obs}}$ and missing parts $Z_{\text{miss}}$. Let $R$ be a binary indicator matrix matching the dimensions of $Z_{\text{com}}$, with elements indicating observed (1) or missing (0) values. The core idea of Rubin (1976) is to treat $R$ as a realization of a random vector following an unknown distribution.

The mechanism is MCAR if $P(R|Z_{\text{com}}) = P(R)$, meaning $R$ is independent of data values, justifying the practice of ignoring missing data. The mechanism is MAR if $P(R|Z_{\text{com}}) = P(R|Z_{\text{obs}})$, where $R$ depends only on observed data, making certain imputation methods viable. If neither condition holds, the mechanism is MNAR. For known missingness mechanisms, specific imputations might be justified (Rubin, 1976; Lachin, 1999). However, with unknown mechanisms, ignoring or imputing data can lead to invalid results, as shown in our simulations in Section 5. Notably, our method does not depend on the missing data values, ensuring valid results regardless of the missingness mechanism. This makes our method particularly suitable for the MNAR case.

**Maximum Mean Discrepancy.** The Maximum Mean Discrepancy (MMD) is a two-sample test used for deciding if two groups of independent observations $X = \{x_1, \ldots, x_{n_1}\}$ and $Y = \{y_1, \ldots, y_{n_2}\}$ are sampled from different distributions $p$ and $q$. Its test statistic is a kernel-based measure of distance which compares the mean embeddings of $p$ and $q$ in a reproducing kernel Hilbert space (Aronszajn, 1950). More formally, let $\mathcal{H}_k$ denote a reproducing kernel Hilbert space with kernel function $k$, the MMD between $p$ and $q$ is defined as (Gretton et al., 2012a) the following statistic (often written as $\text{MMD}^2$):

$$\text{MMD}^2[\mathcal{H}_k, p, q] = \left( \sup_{f \in \mathcal{H}_k: ||f|| \leq 1} (\mathbb{E}_{X \sim p}[f(X)] - \mathbb{E}_{Y \sim q}[f(Y)]) \right)^2.$$

In the context of two-sample testing, an unbiased estimate of $\text{MMD}^2[\mathcal{H}_k, p, q]$ is:

$$\text{MMD}_u^2(X, Y) = \frac{1}{n_1(n_1-1)} \sum_{i=1}^{n_1} \sum_{\substack{j=1 \\ j \neq i}}^{n_2} k(x_i, x_j) + \frac{1}{n_2(n_2-1)} \sum_{i=1}^{n_2} \sum_{\substack{j=1 \\ j \neq i}}^{n_2} k(y_i, y_j) - \frac{2}{n_1 n_2} \sum_{i=1}^{n_1} \sum_{j=1}^{n_2} k(x_i, y_j).$$

The MMD test rejects the null hypothesis when $\text{MMD}_u^2(X, Y)$ exceeds certain threshold, usually chosen to control the probability of Type I error under a pre-specified number $\alpha \in (0, 1)$. It is shown in Gretton et al. (2012a) that when $k$ is a characteristic kernel (Sriperumbudur et al., 2011), $\text{MMD}^2[\mathcal{H}_k, p, q] = 0$ if and only if $p = q$, otherwise $\text{MMD}^2[\mathcal{H}_k, p, q] > 0$. Hence, when a characteristic kernel is used, the MMD test is able to detect any distribution changes. However, on finite sample sizes, the choices of kernel have significant impact on the power of MMD test (Biggs et al., 2023). The Laplacian and Gaussian kernels are defined as

$$k_L(x, y) = \exp(-\beta ||x - y||_1), \text{ and } k_G(x, y) = \exp(-\gamma ||x - y||_2^2),$$

respectively, where $\beta > 0$ and $\gamma > 0$ are hyperparameters.

The Laplacian and Gaussian kernels are popular choices for the MMD (Gao & Shao, 2023; Sriperumbudur et al., 2011), mainly because they are both characteristic kernels (Sriperumbudur et al., 2010; 2011). There are other characteristic kernels, such as the Matérn family of kernels (Sriperumbudur et al., 2010). In the univariate case, the MMD test has been shown to perform similarly for the Laplacian and Gaussian kernels (Bodenham & Kawahara, 2023).

In this work we only use the Laplacian kernel $k_L$ because it allows us to derive appropriate bounds for our method. Using a different kernel with this approach may be possible but would require the derivation of different bounds. While this may seem restrictive, we emphasise that the Laplacian kernel is characteristic and so can be used with the MMD to detect any change in distribution. In the following, we use $k(x, y)$ to denote the Laplacian kernel without causing ambiguity.

**Strengths and limitations of the MMD.** As discussed, the core principle of MMD is to detect differences between two distributions by measuring the largest difference in expectations over a rich class of functions in a reproducing kernel Hilbert space. This makes the MMD capable of capturing a wide range of distributional discrepancies, including subtle differences in higher-order moments or tail behaviour, which classical tests such as Wilcoxon-Mann-Whitney test may overlook. The MMD is compared to popular univariate testing methods in Bodenham & Kawahara (2023). Furthermore, Since the MMD is based on a kernel, it can be applied to high-dimensional data naturally, while the Wilcoxon-Mann-Whitney and Kolmogorov-Smirnov tests are restricted to univariate data. However, a limitation of the MMD test is that its statistical performance depends on the choice of the kernel and the corresponding parameter value for the kernel. Several methods are proposed for dealing with these issues, which are discussed in the related work (Gretton et al., 2012b; Biggs et al., 2023). In this work we restrict the MMD kernel to the Laplacian kernel, which is characteristic and so can be used to detect any change in distribution (Sriperumbudur et al., 2010; 2011), and set the kernel parameter using the median heuristic, which has been shown to generally perform well (Gretton et al., 2012a; Bodenham & Kawahara, 2023).

**Permutation Test.** The permutation test is a common numerical procedure for determining the statistical significance of a computed two-sample test statistic. It relies on the principle that if the null hypothesis of equal distribution is true, the observed values are exchangeable, and so all permutations are equally likely (Good, 2005). The permutation test is non-parametric, meaning that it does not assume the distribution of the data. Furthermore, Schrab et al. (2023) has shown that this test produces the $p$-value non-asymptotically at most the significance level $\alpha$ under the null hypothesis. In contrast, the testing approach based on normal approximations (Gao & Shao, 2023; Shekhar et al., 2022) are only asymptotic true.

To be more specific, let $X = \{x_1, \ldots, x_{n_1}\}$ and $Y = \{y_1, \ldots, y_{n_2}\}$, where each sample is $\mathbb{R}^d$ real value. The permutation test begins with uniformly sampling $B$ i.i.d. permutations from $\{1, \ldots, n_1 + n_2\}$. Let us denote the $B$ samplings as $(\sigma^{(1)}, \ldots, \sigma^{(B)})$, where each $\sigma^{(i)} = (\sigma^{(i)}(1), \ldots, \sigma^{(i)}(n_1 + n_2))$, $i = 1, \ldots, B$. Subsequently, define $(z_1, \ldots, z_{n_1}) = (x_1, \ldots, x_{n_1})$ and $(z_{n_1+1}, \ldots, z_{n_1+n_2}) = (y_1, \ldots, y_{n_2})$, and for any $i = 1, \ldots, B$, denote

$$X_{\sigma^{(i)}} = \{z_{\sigma^{(i)}(1)}, \ldots, z_{\sigma^{(i)}(n_1)}\}, Y_{\sigma^{(i)}} = \{z_{\sigma^{(i)}(n_1+1)}, \ldots, z_{\sigma^{(i)}(n_1+n_2)}\}.$$

The test threshold of MMD test, using $\text{MMD}_u^2(X, Y)$ as test statistic with significance level $0 < \alpha < 1$, is then computed as the $\lceil \alpha(B+1) \rceil$-th largest numbers in the set

$$\left\{\text{MMD}_u^2(X, Y)\right\} \cup \left\{\text{MMD}_u^2(X_{\sigma^{(i)}}, Y_{\sigma^{(i)}}), i = 1, \ldots, B\right\}.$$

If $\text{MMD}_u^2(X, Y)$ exceeds this threshold, then MMD test rejects the null hypothesis. Otherwise, the null hypothesis is not rejected.

**Normal Approximation.** It is well known that for a fixed dimension $d$, the asymptotic distribution of $\text{MMD}_u^2(X, Y)$ under the null hypothesis (i.e. $p = q$) takes the form of an infinite weighted sum of $\chi^2$ random variables with the weights depending on $p$ (Gretton et al., 2012a), which are normally unknown. An estimation of the asymptotic distribution of $\text{MMD}_u^2(X, Y)$ is established in Gretton et al. (2009). This estimation is proved asymptotically correct regardless of the distribution $p$ under certain conditions. However, empirical experiments indicate this estimation is rather conservative.

It was recently shown by Gao & Shao (2023) that the studentized $\text{MMD}_u^2(X, Y)$ given by equation 8 convergences to normal distribution when all $n, m, d \to \infty$. This result holds for a wide range of kernels, including Laplacian and Gaussian kernels. This normal approximation is shown empirically to be rather accurate (Gao & Shao, 2023) even for values of relatively small $n, m, d$, e.g. $n = m = 25, d = 50$, and a higher approximation accuracy is shown for larger values of $n, m, d$.

**Related Work.** Various approaches have been proposed to mitigate the issue of missing data for two-sample hypothesis testing. When the missing data are missing at random, rank-based two-sample testing methods (Cheung, 2005; Chen et al., 2013) have been proposed which control the Type I error and achieve good testing power. Paired two-sample testing with missing data are studied in Martínez-Camblor et al. (2013), and Fong et al. (2018), under the assumption that data are missing completely at random and at least one of the paired samples is observed. When the samples are bounded within an interval, Pan (2000) proposed an algorithm based on multiple imputation (Little & Rubin, 2019) by employing approximate Bayesian bootstrap (Rubin & Schenker, 1986). Shutoh et al. (2010) proposed a method for testing the shift in the mean of high-dimensional normal data assuming the two samples share the same missingness mechanism. By considering certain unobserved samples as the "worst case", Lachin (1999) proposes a test method based on rank called the worst rank test.

All methods above rely on different assumptions about the missing data. One exception of existing methods which makes no missing data assumption, as we mentioned before, is the rank based two-sample testing method proposed in Zeng et al. (2024). Their method rejects the null hypothesis after accounting for all possible Wilcoxon-Mann-Whitney test statistics by deriving the bounds of the test statistics. Our method in this work follows the same approach as in Zeng et al. (2024). We make no assumption of the missing data, except assuming they are distinct. The bounds of the MMD test statistic using the Laplacian kernel are derived, and the null hypothesis is rejected when all possible test statistics are significant. The approach in Zeng et al. (2024) is restricted to univariate data for testing a location shift, since it relies on the Wilcoxon-Mann-Whitney test, while our approach can be used for both univariate and multivariate samples, and can detect any distributional change since it is based on the MMD test.

Our method is deliberately conservative, contrasting to the dominant methods mentioned above, which seek to boost the testing power when certain assumptions of the missing data can be made. Our concern here is whether the testing result is significant when *no* missing data assumption is considered appropriate. One related concept is known as partial identification (Manski, 2003), which argues that it is not sufficient to know what inferences can be made when certain assumptions are employed. It highlights the importance of making inferences when no assumption is imposed for drawing robust conclusions.

The concern that a set of plausible missing data assumptions might lead to different conclusions often motivates the use of sensitivity analysis (Thabane et al., 2013; Mathur, 2023), where these assumptions are evaluated to determine whether a consensus can be reached about the testing conclusions. If all evaluated assumptions lead to a consistent conclusion, then the conclusion is considered to be robust and is taken as the final result, as demonstrated by Bakris et al. (2015) . Our method can be viewed as a sensitivity analysis that incorporates all possible assumptions. If a significant testing result is determined by our method, then the result is significant regardless of the values of the missing data, and hence is robust to all possible assumptions on the missing data.

Various authors have proposed extensions of the MMD two-sample test for several different purposes. Gretton et al. (2012b) and Biggs et al. (2023) investigated strategies for optimal kernel selection based on the observed data, while Schrab et al. (2023) focused on selecting an appropriate bandwidth for the chosen kernel. To handle complex data modalities such as images, audio, and video in two sample testing, Kirchler et al. (2020) proposed a deep learning-based extension of the MMD test. As we will discuss below, a common implementation of the MMD test relies on a computationally intensive procedure known as permutation testing to approximate the null distribution and compute a *p*-value. To alleviate the computational burden associated with the permutation procedure, several works (Shekhar et al., 2022; Gao & Shao, 2023) have developed permutation-free approaches that significantly reduce computation. Our research in this work explores a different direction by studying MMD-based two-sample testing in the presence of missing data. We propose tests that control the Type I error without requiring assumptions about the underlying missingness mechanism.

## 3 Bounding MMD with Missing Data

In this section, we provide bounds for the MMD test statistic in the presence of missing data. As mentioned in the Section 1, the key idea of the proposed method is to consider all possible test statistics in the presence

of missing data and reject the null hypothesis only when all test statistics are significant. To compute these bounds, we use the Laplacian kernel, which is a characteristic kernel (Sriperumbudur et al., 2010; 2011) and able to detect any change in distribution.

**Univariate Data.** If a subset of data in $X$ and $Y$ are unobserved, the $\mathrm{MMD}_u^2(X, Y)$ test statistic cannot be directly computed. However, the test statistic can be bounded given the missing data. Notice that for any two given real values $x$ and $y$, the Laplacian kernel $k(x, y) = \exp\left(-\beta |x - y|\right)$ is bounded by $(0, 1]$ since $\beta > 0$ and $|x - y| > 0$. Hence, a straightforward way to obtain a lower bound of $\mathrm{MMD}_u^2(X, Y)$ is to let $k(x_i, x_j) = 0$ if at least one value of $x_i$ and $x_j$ is unobserved. Similarly, if at least one value of $y_i$ and $y_j$ is unobserved, let $k(y_i, y_j) = 0$. On the other hand, to minimize $-k(x_i, y_j)$, we can let $-k(x_i, y_j) = -1$ if either $x_i$ or $y_j$ is unobserved. The upper bound of $\mathrm{MMD}_u^2(X, Y)$ can be obtained following a similar manner. This simplistic method, while providing a theoretically correct bound of $\mathrm{MMD}_u^2(X, Y)$, might yield bounds that are too conservative for effective two-sample testing in practice.

In the following, we construct tighter bounds of $\mathrm{MMD}_u^2(X, Y)$ in order to make it useful for the two-sample testing problem. One starts by decomposing $\mathrm{MMD}_u^2(X, Y)$ into terms which have either none, one or both arguments of the kernel function missing. Without loss of generality, let us assume $x_1, \ldots, x_{m_1}, y_1, \ldots, y_{m_2}$ are not observed, or not fully observed, while $x_{m_1+1}, \ldots, x_{n_1}$ and $y_{m_2+1}, \ldots, y_{n_2}$ are fully observed. Then, $\mathrm{MMD}_u^2(X, Y)$ can be decomposed, using the following lemma:

**Lemma 1.** *Suppose* $X = \{x_1, \ldots, x_{n_1}\}$, $Y = \{y_1, \ldots, y_{n_2}\}$ *are subsets of* $\mathbb{R}^d$, *where* $d \geq 1$. *Suppose that* $MMD_u^2(X, Y)$ *is unbiased MMD test statistic defined as*

$$MMD_u^2(X, Y) = \frac{1}{n_1(n_1-1)} \sum_{i=1}^{n_1} \sum_{\substack{j=1 \\ j \neq i}}^{n_1} k(x_i, x_j) + \frac{1}{n_2(n_2-1)} \sum_{i=1}^{n_2} \sum_{\substack{j=1 \\ j \neq i}}^{n_2} k(y_i, y_j) - \frac{2}{n_1 n_2} \sum_{i=1}^{n_1} \sum_{j=1}^{n_2} k(x_i, y_j), \qquad (1)$$

*with* $k$ *denoting the Laplacian kernel. Then, for any two positive integers* $m_1$, $m_2$ *such that* $m_1 \leq n_1$, $m_2 \leq n_2$, $MMD_u^2(X, Y)$ *can be rewritten as:*

$$MMD_u^2(X, Y) = A_1 + A_2 + A_3 + A_4,$$

*where* $c_1 = \frac{2}{n_1(n_1-1)}$, $c_2 = \frac{2}{n_2(n_2-1)}$, $c_3 = \frac{2}{n_1 n_2}$, *and*

$$A_1 = c_1 \sum_{i=1}^{m_1} \sum_{j=i+1}^{m_1} k(x_i, x_j) + c_2 \sum_{i=1}^{m_2} \sum_{j=i+1}^{m_2} k(y_i, y_j) - c_3 \sum_{i=1}^{m_1} \sum_{j=1}^{m_2} k(x_i, y_j),$$

$$A_2 = c_1 \sum_{i=m_1+1}^{n_1-1} \sum_{j=i+1}^{n_1} k(x_i, x_j) + c_2 \sum_{i=m_2+1}^{n_2-1} \sum_{j=i+1}^{n_2} k(y_i, y_j) - c_3 \sum_{i=m_1+1}^{n_1} \sum_{j=m_2+1}^{n_2} k(x_i, y_j),$$

$$A_3 = c_1 \sum_{i=1}^{m_1} \sum_{j=m_1+1}^{n_1} k(x_i, x_j) - c_3 \sum_{i=1}^{m_1} \sum_{j=m_2+1}^{n_2} k(x_i, y_j),$$

$$A_4 = c_2 \sum_{i=1}^{m_2} \sum_{j=m_2+1}^{n_2} k(y_i, y_j) - c_3 \sum_{i=m_1+1}^{n_1} \sum_{j=1}^{m_2} k(x_i, y_j).$$

The proof of this lemma can be found in Appendix A.1.

Lemma 1 decomposes $\mathrm{MMD}_u^2(X, Y)$ into four parts: the first part $A_1$ including only the unobserved data in $X$ and $Y$; the second part $A_2$ including only the fully observed data in $X$ and $Y$; the third and the fourth terms $A_3$ and $A_4$ involving data where one sample (either $X$ or $Y$) contains only unobserved data while the other sample includes only fully observed data.

In order to bound $\mathrm{MMD}_u^2(X, Y)$, we propose to bound the four terms $A_1, A_2, A_3$ and $A_4$ separately. Given that $A_2$ includes the fully observed data only, it is thus sufficient to bound $A_1, A_3$ and $A_4$. For the first term $A_1$, where all data are unobserved, or not fully observed, it can be seen that $1 \geq k(x, y) > 0$ for any $x, y \in \mathbb{R}$.

Hence, we have $m_1(m_1-1)/2 \geq \sum_{i=1}^{m_1} \sum_{j=i+1}^{m_1} k(x_i, x_j) > 0$, $m_2(m_2-1)/2 \geq \sum_{i=1}^{m_2} \sum_{j=i+1}^{m_2} k(y_i, y_j) > 0$, and $m_1 m_2 \geq \sum_{i=1}^{m_1} \sum_{j=1}^{m_2} k(x_i, y_j) > 0$. Then according to the definition of $A_1$ in Lemma 1, we have

$$\frac{m_1(m_1-1)}{n_1(n_1-1)} + \frac{m_2(m_2-1)}{n_2(n_2-1)} > A_1 > -\frac{2}{n_1 n_2} m_1 m_2. \tag{2}$$

The challenge of bounding $\mathrm{MMD}_u^2(X, Y)$ is to provide tight bounds for terms $A_3$ and $A_4$, where data can be fully observed in one sample and missing, or partially observed in another sample. We focus on the term $A_3$ because $A_4$ can be bounded similarly. The key to bounding $A_3$ is by first rewriting $A_3$ as

$$A_3 = \sum_{i=1}^{m_1} \left( c_1 \sum_{j=m_1+1}^{n_1} k(x_i, x_j) - c_3 \sum_{j=m_2+1}^{n_2} k(x_i, y_j) \right), \tag{3}$$

where $c_1 = \frac{2}{n_1(n_1-1)}$ and $c_3 = \frac{2}{n_1 n_2}$. Define $T_1(z)$ as

$$T_1(z) = c_1 \sum_{j=m_1+1}^{n_1} k(z, x_j) - c_3 \sum_{j=m_2+1}^{n_2} k(z, y_j).$$

Then, we have

$$A_3 = \sum_{i=1}^{m_1} T_1(x_i),$$

i.e. $A_3$ is the summation of the function values $T_1(z)$ applied to all unobserved samples in $X$. This equation suggests that in order to bound $A_3$, it is sufficient to bound the values of $T_1(z)$. Therefore, we proceed by bounding $T_1(z)$, using the following result for univariate data:

**Lemma 2.** *Let $x_1, \ldots, x_{\ell_1}$ and $y_1, \ldots, y_{\ell_2}$ be real values, which are observed. Suppose $a_1, \ldots, a_{\ell_1}$, $b_1, \ldots, b_{\ell_2}$ and $\beta$ are positive constants. Define*

$$T(z) = \sum_{i=1}^{\ell_1} a_i \exp(-\beta|x_i - z|) - \sum_{i=1}^{\ell_2} b_i \exp(-\beta|y_i - z|) \tag{4}$$

*as a function of $z \in \mathbb{R}$. Subsequently, for any given $z_0 \in \mathbb{R}$, we have*

$$T(z_0) \geq \min\{0, T(x_1), \ldots, T(x_{\ell_1}), T(y_1), \ldots, T(y_{\ell_2})\},$$

*and*

$$T(z_0) \leq \max\{0, T(x_1), \ldots, T(x_{\ell_1}), T(y_1), \ldots, T(y_{\ell_2})\}.$$

*Moreover, these bounds for $T(z_0)$ are tight, and we have equality when $z_0 \in \{x_1, \ldots, x_{\ell_1}, y_1, \ldots, y_{\ell_2}\}$ or as $z_0 \to \pm\infty$.*

This lemma is proved in Appendix A.2. It provides a linear time algorithm for computing the bounds of $T(z)$ by computing all the values in the set $\{T(z) : z \in \{x_1, \ldots, x_{\ell_1}, y_1, \ldots, y_{\ell_2}\}\}$ and identifying the maximum and minimum values.

After proving Lemma 2, we can provide the bounds of $\mathrm{MMD}_u^2(X, Y)$ in the presence of missing data, without making missing data assumptions.

**Theorem 1.** *Suppose $X = \{x_1, \ldots, x_{n_1}\}$ and $Y = \{y_1, \ldots, y_{n_2}\}$ are univariate real values. Assume $x_1, \ldots, x_{m_1}, y_1, \ldots, y_{m_2}$ are unobserved. Let $k$ denote the Laplacian kernel and define*

$$T_1(z) = c_1 \sum_{j=m_1+1}^{n_1} k(z, x_j) - c_3 \sum_{j=m_2+1}^{n_2} k(z, y_j),$$

$$T_2(z) = c_2 \sum_{j=m_2+1}^{n_2} k(z, x_j) - c_3 \sum_{j=m_1+1}^{n_1} k(z, y_j),$$

where $c_1 = \frac{2}{n_1(n_1-1)}, c_2 = \frac{2}{n_2(n_2-1)}$ and $c_3 = \frac{2}{n_1 n_2}$. Further, let

$$S_1 = \{0, T_1(x_{m_1+1}), \ldots, T_1(x_{n_1}), T_1(y_{m_2+1}), \ldots, T_1(y_{n_2})\},$$
$$S_2 = \{0, T_2(x_{m_2+1}), \ldots, T_2(x_{n_2}), T_2(y_{m_1+1}), \ldots, T_2(y_{n_1})\}.$$

Then, $MMD_u^2(X,Y)$ defined in equation 1 using Laplacian kernel $k$ is bounded, over all possible values of $x_1, \ldots, x_{m_1}, y_1, \ldots, y_{m_2}$, as follows:

$$\frac{m_1(m_1-1)}{n_1(n_1-1)} + \frac{m_2(m_2-1)}{n_2(n_2-1)} + m_1 \max S_1 + m_2 \max S_2 + A_2 \geq MMD_u^2(X,Y),$$

$$MMD_u^2(X,Y) \geq A_2 + m_1 \min S_1 + m_2 \min S_2 - \frac{2}{n_1 n_2} m_1 m_2,$$

where $A_2$ is defined in lemma 1.

The proof of this theorem is provided in Appendix A.5.

**Multivariate Data.** We now provide the bounds of $\text{MMD}_u^2(X,Y)$ for multivariate data, where $X = \{x_1, \ldots, x_{n_1}\}$, $Y = \{y_1, \ldots, y_{n_2}\}$ and $X, Y \in \mathbb{R}^d$ with $d > 1$. For each $x_i$ in $X$, denote $x_i = (x_i(1), \ldots, x_i(d))$, where $x_i(l) \in \mathbb{R}$ is the value the $l$th component of $x_i$. Similarly, for each $y_i$ in $Y$, denote $y_i = (y_i(1), \ldots, y_i(d))$. Note that $X$ and $Y$ are assumed to have the same dimension $d$.

As in the case for univariate data, one starts by decomposing $\text{MMD}_u^2(X,Y)$ into $A_1, A_2, A_3$ and $A_4$ using Lemma 1. Term $A_2$ only includes data that are fully observed. Thus it can be computed directly. Term $A_1$ can be bounded using simple inequalities, as shown in equation 23 in the Appendix. The challenge is to provide tight bounds for term $A_3$ and $A_4$.

To bound term $A_3$ and $A_4$, we first provide notation for which components of a $d$-dimensional observation are missing and define what it means to impute such a value.

**Definition 1.** *For a value $z = (z(1), \ldots, z(d)) \in \mathbb{R}^d$, let $U_z \subset \{1, 2, \ldots, d\}$ denote the set of components of $z$ that are missing; in other words $\{z(j) : j \in U_z\}$ are missing.*

**Definition 2.** *For a value $z \in \mathbb{R}^d$, that has missing components, $z^*$ is called an imputation of $z$ if $z^* \in \mathbb{R}^d$ is fully observed and $z^*(j) = z(j)$ for all $j \in \{1, \ldots, d\} \setminus U_z$.*

We can now extend Lemma 2 to the $d$-dimensional case:

**Lemma 3.** *Let $x_1, \ldots, x_{\ell_1}, y_1, \ldots, y_{\ell_2} \in \mathbb{R}^d$ be values that are fully observed. Suppose $a_1, \ldots, a_{\ell_1}, b_1, \ldots, b_{\ell_2}, \beta$ are positive constants. For $z = (z(1), \ldots, z(d)) \in \mathbb{R}^d$ with missing components, define*

$$T(\{z(j) : j \in U_z\}) = \sum_{i=1}^{\ell_1} a_i \exp\left(-\beta \sum_{j \in U_z} |x_i(j) - z(j)|\right) - \sum_{i=1}^{\ell_2} b_i \exp\left(-\beta \sum_{j \in U_z} |y_i(j) - z(j)|\right)$$

*as a function of the unobserved components of $z$ and let*

$$\mathcal{X} = \{T(\{z(j) : j \in U_z\}) \; : \; z(i) \in \{x_1(i), \ldots, x_{\ell_1}(i), y_1(i), \ldots, y_{\ell_2}(i)\}, i \in U_z\}.$$

*Then, for any possible imputation $z^*$ of $z$,*

$$\min\{0, \min \mathcal{X}\} \leq T(\{z^*(j) : j \in U_z\}) \leq \max\{0, \max \mathcal{X}\}$$

Lemma 3 is proved in Appendix A.3. It shows that in order to compute the maximum and minimum values of $T(\{z(j) : j \in U_z\})$ for $z \in \mathbb{R}^d$, it is sufficient to check the imputations of $z$ where its missing components are imputed using the components of $x_1, \ldots, x_{\ell_1}, y_1, \ldots, y_{\ell_2}$. However, computing $T(\{z(j) : j \in U_z\})$ for all possible imputations using Lemma 3 is $(\ell_1 + \ell_2)^{|U_{x_i}|}$, which is exponential in the number of unobserved components of $z$ and impractical to compute. To address this computational challenge, we propose to further bound $\min \mathcal{X}$ and $\max \mathcal{X}$, using the following result:

**Lemma 4.** *Following the notation and definitions in Lemma 3, denote*

$$\tilde{x}_i(j) = \max\{|x_i(j) - x_1(j)|, \ldots, |x_i(j) - x_{\ell_1}(j)|, |x_i(j) - y_1(j)|, \ldots, |x_i(j) - y_{\ell_2}(j)|\}$$

*for any $i \in \{1, \ldots, \ell_1\}, j \in U_z$, and denote*

$$\tilde{y}_i(j) = \max\{|y_i(j) - x_1(j)|, \ldots, |y_i(j) - x_{n_1}(j)|, |y_i(j) - y_1(j)|, \ldots, |y_i(j) - y_{\ell_2}(j)|\}$$

*for any $i \in \{1, \ldots, \ell_2\}, j \in U_z$. Subsequently, we have*

$$\min \mathcal{X} \geq \sum_{i=1}^{\ell_1} a_i \exp\left(-\beta \sum_{j \in U_z} \tilde{x}_i(j)\right) - \sum_{i=1}^{\ell_2} b_i, \ \max \mathcal{X} \leq \sum_{i=1}^{\ell_1} a_i - \sum_{i=1}^{\ell_2} b_i \exp\left(-\beta \sum_{j \in U_z} \tilde{y}_i(j)\right).$$

This lemma is proved in Appendix A.4, it provides the lower and upper bounds for $\min \mathcal{X}$ and $\max \mathcal{X}$ defined in Lemma 3.

The bounds given by Lemma 4 can be computed efficiently. For each $i \in \{1, \ldots, \ell_1\}$ and $j \in U_z$, $|\tilde{x}_i(j)| = \ell_1 + \ell_2$ according to its definition in Lemma 4. Hence, in order to bound $\min \mathcal{X}$, computational times $|U_z|\ell_1(\ell_1 + \ell_2)$ are needed, which is of order $O(dn_1(n_1 + n_2))$. Similarly, to bound $\max \mathcal{X}$, the order of computation complexity is $O(dn_2(n_1 + n_2))$. This computational cost reduction, compared with Lemma 3, which grows exponentially with $|U_z|$, makes bounding $\text{MMD}_u^2(X, Y)$ possible in practice.

We are now ready to prove the main result for the multivariate data. In order to state the result, we first make the following two definitions. The first definition defines incomplete Laplacian kernel, which allows us to compute the Laplacian distance between two partially observed data points.

**Definition 3.** *For any $x, y \in \mathbb{R}^d$, let $U_{x,y} \subset \{1, \ldots, d\}$ be the index of dimensions for which either $x$ or $y$ have components that are not observed. Let $k$ denote the Laplacian kernel with parameter $\beta$. Then, the incomplete Laplacian kernel is defined as*

$$\mathcal{K}(x, y) = \exp\left(-\beta \sum_{i \in \{1, \ldots, d\} \setminus (U_x \cup U_y)} |x(i) - y(i)|\right).$$

The second definition defines constants $c_1, c_2, c_3$ and $C_1, C_2, C_3, C_4$ depending on the observed data for bounding the $\text{MMD}_u^2(X, Y)$ test statistic.

**Definition 4.** *Suppose $X = \{x_1, \ldots, x_{n_1}\}$ and $Y = \{y_1, \ldots, y_{n_2}\}$ are samples of $\mathbb{R}^d$ real values. Assume $x_1, \ldots, x_{m_1}$, $y_1, \ldots, y_{m_2}$ are data that are missing, or not completely observed. Let $k$ denote the Laplacian kernel and $\mathcal{K}$ denote the incomplete Laplacian kernel defined in Definition 3. Further, denote*

$$\tilde{x}_i(l) = \max\{|x_i(l) - x_{m_1+1}(l)|, \ldots, |x_i(l) - x_{n_1}(l)|, |x_i(l) - y_{m_2+1}(l)|, \ldots, |x_i(l) - y_{n_2}(l)|\}$$

*for any $i \in \{m_1 + 1, \ldots, n_1\}, l \in \{1, \ldots, d\}$, and denote*

$$\tilde{y}_i(l) = \max\{|y_i(l) - x_{m_1+1}(l)|, \ldots, |y_i(l) - x_{n_1}(l)|, |y_i(l) - y_{m_2+1}(l)|, \ldots, |y_i(l) - y_{n_2}(l)|\}$$

*for any $i \in \{m_2 + 1, \ldots, n_2\}, l \in \{1, \ldots, d\}$. Let*

$$C_1 = \sum_{i=1}^{m_1} \max\left\{0, c_1 \sum_{j=m_1+1}^{n_1} \mathcal{K}(x_i, x_j) - c_3 \sum_{j=m_2+1}^{n_2} \mathcal{K}(x_i, y_j) \exp\left(-\beta \sum_{l \in U_{x_i}} \tilde{y}_j(l)\right)\right\},$$

$$C_2 = \sum_{i=1}^{m_1} \min\left\{0, c_1 \sum_{j=m_1+1}^{n_1} \mathcal{K}(x_i, x_j) \exp\left(-\beta \sum_{l \in U_{x_i}} \tilde{x}_j(l)\right) - c_3 \sum_{j=m_2+1}^{n_2} \mathcal{K}(x_i, y_j)\right\},$$

$$C_3 = \sum_{i=1}^{m_2} \max\left\{0, c_2 \sum_{j=m_2+1}^{n_2} \mathcal{K}(y_i, y_j) - c_3 \sum_{j=m_1+1}^{n_1} \mathcal{K}(y_i, x_j) \exp\left(-\beta \sum_{l \in U_{y_i}} \tilde{x}_j(l)\right)\right\},$$

$$C_4 = \sum_{i=1}^{m_2} \min\left\{0, c_2 \sum_{j=m_2+1}^{n_2} \mathcal{K}(y_i, y_j) \exp\left(-\beta \sum_{l \in U_{y_i}} \tilde{y}_j(l)\right) - c_3 \sum_{j=m_1+1}^{n_1} \mathcal{K}(y_i, x_j)\right\},$$

*where $c_1 = \frac{2}{n_1(n_1-1)}, c_2 = \frac{2}{n_2(n_2-1)}$ and $c_3 = \frac{2}{n_1 n_2}$.*

Then, using Definition 3 and Definition 4, the main result can be stated as

**Theorem 2.** *Suppose $X = \{x_1, \ldots, x_{n_1}\}$ and $Y = \{y_1, \ldots, y_{n_2}\}$ are samples of $\mathbb{R}^d$ real values. Assume $x_1, \ldots, x_{m_1}, y_1, \ldots, y_{m_2}$ are data that are missing, or not completely observed. Subsequently, the $MMD_u^2(X, Y)$ using Laplacian kernel $k$ is bounded, over all possible imputations of $x_1, \ldots, x_{m_1}, y_1, \ldots, y_{m_2}$, as follows:*

$$c_1 \sum_{i=1}^{m_1} \sum_{j=i+1}^{m_1} \mathcal{K}(x_i, x_j) + c_2 \sum_{i=1}^{m_2} \sum_{j=i+1}^{m_2} \mathcal{K}(y_i, y_j) + A_2 + C_1 + C_3 > MMD_u^2(X, Y),$$

$$MMD_u^2(X, Y) > -c_3 \sum_{i=1}^{m_1} \sum_{j=1}^{m_2} \mathcal{K}(x_i, y_j) + A_2 + C_2 + C_4,$$

*where $\mathcal{K}$ is the incomplete Laplacian kernel defined in Definition 3, $A_2$ is a constant of observed data defined in Lemma 1, and $c_1, c_2, c_3$ and $C_1, C_2, C_3, C_4$ are also constants of the observed data defined in Definition 4.*

The proof of this theorem is included in Appendix A.6.

**Remark 1.** As we discussed previously, our strategy for bounding the MMD test statistic $\mathrm{MMD}_u^2(X, Y)$ is to provide the bounds for the terms $A_1$, $A_2$, $A_3$ and $A_4$, separately. Since $A_1$ includes only the unobserved data in $X$ and $Y$, the bounds of it can be constructed according to Inequality (2) and cannot be improved. The second term $A_2$ includes only observed data. The third and the fourth term $A_3$, $A_4$ can be decomposed into the sum of functions of $T(x)$, as shown in Inequality (3). We show in Lemma 2 that the tight lower and upper bounds of $T(x)$ can be constructed. Hence the bounds of term $A_3$ and $A_4$ are tight. While all the bounds of $A_1$, $A_2$, $A_3$ and $A_4$ are tight, our current bounds for the MMD test statistic are not tight since we provide the bounds of $A_1$, $A_2$, $A_3$, $A_4$ separately, rather than jointly.

## 4 Two-Sample Testing in the Presence of Arbitrarily Missing Data

In this section, we discuss methods for employing bounds derived in Section 3 to develop valid two-sample testing methods. Besides computing the test statistic, we also need a method for computing a $p$-value from the test statistic.

**Bounding $p$-value using permutations.** The permutation test, as discussed in Section 2, is a numerical procedure that can be used to compute a $p$-value for a test statistic based on the observed data. The $p$-value of the MMD test using permutations is computed as

$$p = \frac{1}{B+1} \left( 1 + \sum_{i=1}^{B} \mathbb{I}(\mathrm{MMD}_u^2(X_{\sigma^{(i)}}, Y_{\sigma^{(i)}}) \geq \mathrm{MMD}_u^2(X, Y)) \right), \tag{5}$$

where $\mathbb{I}(E)$ is the indicator function such that if the event $E$ occurs, $\mathbb{I}(E) = 1$, otherwise $\mathbb{I}(E) = 0$. The null hypothesis is then rejected if $p$ is smaller or equal to the pre-specified significance level $\alpha$.

Using the bounds of the MMD in the presence of missing data, we proceed by providing bounds of the $p$-value, using the following result:

**Theorem 3.** *Suppose $X = \{x_1, \ldots, x_{n_1}\}, Y = \{y_1, \ldots, y_{n_2}\} \in \mathbb{R}^d$. Let $(\sigma^{(1)}, \ldots, \sigma^{(B)})$ be $B$ i.i.d. random permutations of $\{1, \ldots, n_1 + n_2\}$ and denoted as $\sigma^{(i)} = (\sigma^{(i)}(1), \ldots, \sigma^{(i)}(n_1 + n_2))$, $i = 1, \ldots, B$. Let $z_1 = x_1, \ldots, z_{n_1} = x_{n_1}, z_{n_1+1} = y_1, \ldots, z_{n_1+n_2} = y_{n_2}$ and for any $i = 1, \ldots, B$, denote*

$$X_{\sigma^{(i)}} = \{z_{\sigma^{(i)}(1)}, \ldots, z_{\sigma^{(i)}(n_1)}\}, Y_{\sigma^{(i)}} = \{z_{\sigma^{(i)}(n_1+1)}, \ldots, z_{\sigma^{(i)}(n_1+n_2)}\}.$$

*Define $p$ according to equation 5. Suppose further we have $\underline{MMD}_u^2(X, Y)$ and $\overline{MMD}_u^2(X, Y)$ such that*

$$\underline{MMD}_u^2(X, Y) \leq MMD_u^2(X, Y), \ MMD_u^2(X_{\sigma^{(i)}}, Y_{\sigma^{(i)}}) \leq \overline{MMD}_u^2(X_{\sigma^{(i)}}, Y_{\sigma^{(i)}}), \ i = 1, \ldots, B.$$

*Define*

$$\overline{p} = \frac{1}{B+1} \left( 1 + \sum_{i=1}^{B} \mathbb{I}(\overline{MMD}_u^2(X_{\sigma^{(i)}}, Y_{\sigma^{(i)}}) \geq \underline{MMD}_u^2(X, Y)) \right).$$

*Then, we have $\overline{p} \geq p$.*

The proof of this theorem is included in Appendix A.7. In the presence of missing data, $\mathrm{MMD}_u^2(X, Y)$ cannot be computed directly. However, as shown in Section 3, the lower and upper bounds of $\mathrm{MMD}_u^2(X, Y)$ can be computed when the Laplacian kernel is used. Then, Theorem 3 can be applied to compute $\overline{p}$, the upper bound of $p$-value. It is proved in Schrab et al. (2023) that the MMD test using the $p$-value defined in equation 5 controls the Type I error. Therefore, using the upper bound $\overline{p}$ defined in Theorem 3 also controls the Type I error, since the bounds were derived without any assumptions about the missing data mechanisms.

**Bounding $p$-value using normality approximation.** We have considered how to bound the $p$-value when it is computed using permutations. When the values of $n_1, n_2, d$ are large enough, an alternative method to compute a $p$-value is to use the normality approximation (Gao & Shao, 2023). It is suggested in Gao & Shao (2023) that this approximation is effective when $n_1, n_2 \geq 25, d \geq 50$.

To more formally describe the normality approximation proposed in Gao & Shao (2023), we will follow their notation and definitions in the following of this section. Let us denote the sample sizes for $X$ and $Y$ as $n$ and $m$, respectively, rather than $n_1, n_2$. Denote $n + m = N$, and let $k$ denote the Laplacian kernel. For any $1 \leq s, t \leq N$, define

$$a_{s,t}^k = \begin{cases} k(X_s, X_t) & 1 \leq s, t \leq n, \\ k(X_s, Y_{t-n}) & 1 \leq s \leq n < t \leq N, \\ k(X_t, Y_{s-n}) & 1 \leq t \leq n < s \leq N, \\ k(Y_{s-n}, Y_{t-n}) & n+1 < s, t \leq N, \end{cases}$$

and $A_{s,t}^{k*} = a_{s,t}^k - a_{\cdot,t}^k - a_{s,\cdot}^k + a_{\cdot,\cdot}^k$, where

$$a_{\cdot,t}^k = \frac{1}{N-2} \sum_{i=1}^{N} a_{i,t}^k, \ a_{s,\cdot}^k = \frac{1}{N-2} \sum_{j=1}^{N} a_{s,j}^k, \ a_{\cdot,\cdot}^k = \frac{1}{(N-1)(N-2)} \sum_{i,j=1}^{N} a_{i,j}^k.$$

Further, define

$$\mathcal{V}_{n,m}^{k*} = \frac{1}{N(N-3)} \sum_{s \neq t} \left( A_{s,t}^{k*} \right)^2 - \frac{1}{(N-1)(N-3)}, \tag{6}$$

and

$$c_{n,m} = \frac{2}{n(n-1)} + \frac{4}{nm} + \frac{2}{m(m-1)}. \tag{7}$$

Then, it is proved in Gao & Shao (2023) that the studentized $\mathrm{MMD}_u^2(X, Y)$, taking the form as

$$\frac{\mathrm{MMD}_u^2(X, Y)}{\sqrt{c_{n,m} \mathcal{V}_{n,m}^{k*}}}, \tag{8}$$

converges to a standard normal distribution under the null hypothesis with some conditions (see Theorem 16 in Gao & Shao (2023)). In order to use this result, it is necessary to compute an estimation of variance of $\mathrm{MMD}_u^2(X, Y)$, i.e. $c_{n,m} \mathcal{V}_{n,m}^{k*}$, which can not be computed directly with missing data. To overcome this problem, we propose to also bound $\mathcal{V}_{n,m}^{k*}$. We begin by making the following definitions.

**Definition 5.** *Suppose* $X = \{X_1, \ldots, X_{n_1}\}, Y = \{Y_1, \ldots, Y_{n_2}\} \in \mathbb{R}^d$, *and assume* $X_1, \ldots, X_{m_1}$ *and* $Y_1, \ldots, Y_{m_2}$ *are not observed. Let* $\mathcal{K}$ *denote the incomplete Laplacian kernel defined in Definition 3, and for* $1 \leq s, t \leq N$, *denote*

$$\overline{a}_{s,t}^{\mathcal{K}} = \begin{cases} \mathcal{K}(X_s, X_t) & 1 \leq s, t \leq n, \\ \mathcal{K}(X_s, Y_{t-n}) & 1 \leq s \leq n < t \leq N, \\ \mathcal{K}(X_t, Y_{s-n}) & 1 \leq t \leq n < s \leq N, \\ \mathcal{K}(Y_{s-n}, Y_{t-n}) & n+1 < s, t \leq N, \end{cases} \quad \underline{a}_{s,t}^{\mathcal{K}} = \begin{cases} 0 & 1 \leq s \leq m_1, 1 \leq t \leq N, \\ 0 & 1 \leq s \leq N, n+1 \leq t \leq n + m_2, \\ a_{s,t}^{\mathcal{K}} & otherwise. \end{cases}$$

*Let* $\overline{A}_{s,t}^{\mathcal{K}*} = \overline{a}_{s,t}^{\mathcal{K}} - \underline{a}_{\cdot,t}^{\mathcal{K}} - \underline{a}_{s,\cdot}^{\mathcal{K}} + \overline{a}_{\cdot,\cdot}^{\mathcal{K}}$, *where*

$$\underline{a}_{\cdot,t}^{\mathcal{K}} = \frac{1}{N-2} \sum_{i=1}^{N} \underline{a}_{i,t}^{\mathcal{K}}, \quad \underline{a}_{s,\cdot}^{\mathcal{K}} = \frac{1}{N-2} \sum_{j=1}^{N} \underline{a}_{s,j}^{\mathcal{K}}, \quad \overline{a}_{\cdot,\cdot}^{\mathcal{K}} = \frac{1}{(N-1)(N-2)} \sum_{i,j=1}^{N} \overline{a}_{i,j}^{\mathcal{K}}.$$

*Define*

$$\overline{\mathcal{V}}_{n,m}^{\mathcal{K}*} = \frac{1}{N(N-3)} \sum_{s \neq t} \left( \overline{A}_{s,t}^{\mathcal{K}*} \right)^2 - \frac{1}{(N-1)(N-3)}.$$

Then, we bound $\mathcal{V}_{n,m}^{k*}$ defined in equation 6, and the test statistic defined in equation 8, using the following results.

**Theorem 4.** *Suppose* $X = \{X_1, \ldots, X_{n_1}\}, Y = \{Y_1, \ldots, Y_{n_2}\} \in \mathbb{R}^d$, *and assume* $X_1, \ldots, X_{m_1}$ *and* $Y_1, \ldots, Y_{m_2}$ *are not observed. Let* $\mathcal{K}$ *denote the incomplete Laplacian kernel defined in Definition 3. Then, we have* $\overline{\mathcal{V}}_{n,m}^{\mathcal{K}*} \geq \mathcal{V}_{n,m}^{k*}$, *Further, suppose we have* $\underline{MMD}_u^2(X,Y)$ *such that*

$$0 \leq \underline{MMD}_u^2(X,Y) \leq MMD_u^2(X,Y),$$

*we have*

$$\frac{\underline{MMD}_u^2(X,Y)}{\sqrt{c_{n,m} \overline{\mathcal{V}}_{n,m}^{\mathcal{K}*}}} \leq \frac{MMD_u^2(X,Y)}{\sqrt{c_{n,m} \mathcal{V}_{n,m}^{k*}}},$$

*where* $c_{n,m}$ *and* $\mathcal{V}_{n,m}^{k*}$ *are defined in equation 6, and equation 7, separately, and* $\overline{\mathcal{V}}_{n,m}^{\mathcal{K}*}$ *is defined in Definition 5.*

The proof of this Theorem is included in the Appendix A.8.

In Gao & Shao (2023), the $p$-value is defined as $p = 1 - \Phi \left( \frac{\mathrm{MMD}_u^2(X,Y)}{\sqrt{c_{n,m} \mathcal{V}_{n,m}^{k*}}} \right)$, where $\Phi$ denotes the cumulative density function of the standard normal distribution. In the presence of missing data, we define

$$\overline{p} = \begin{cases} 1 - \Phi \left( \frac{\underline{\mathrm{MMD}}_u^2(X,Y)}{\sqrt{c_{n,m} \overline{\mathcal{V}}_{n,m}^{\mathcal{K}*}}} \right), & \text{if } \underline{\mathrm{MMD}}_u^2(X,Y) \geq 0, \\ 1 & \text{otherwise.} \end{cases}$$

Then, according to Theorem 4, we have $\overline{p} \geq p$. Hence, rejecting the null hypothesis when $\overline{p} < \alpha$, that is, when $\overline{p}$ is smaller than the significance level, will control the Type I error, provided that the $p$-value $p$ defined in Gao & Shao (2023) is valid. This follows because $\overline{p} < \alpha$ implies $p < \alpha$.

**Applying MMD-Miss.** Now we give details of applying our proposed method, namely MMD-Miss, using the results we have shown in Section 3 and Section 4.

In the presence of missing data, the bounds of $\mathrm{MMD}_u^2(X,Y)$ can be computed using the results in Theorem 1, and Theorem 2 when $d = 1$ and $d > 1$, respectively. Provided the bounds of $\mathrm{MMD}_u^2(X,Y)$, Theorem 3 gives a non-trivial upper bound of $p$-value of $\mathrm{MMD}_u^2(X,Y)$ when the permutation test is used, and Theorem 4 gives an upper bound of $p$-value when the normality approximation is used.

Denoting the upper bound of $p$-value as $\overline{p}$, MMD-Miss rejects the null hypothesis when $\overline{p}$ is larger than or equal to the pre-specified significance level $\alpha$. Further, we refer to the MMD-Miss that computes $\overline{p}$ using the results in Theorem 3 with permutation tests as **MMD-Miss-Perm**, and the MMD-Miss that computes $\overline{p}$ using the results in Theorem 4 with normality approximation as **MMD-Miss-Normality**.

We note that, since MMD-Miss-Normality is relied on the normality approximation proposed in Gao & Shao (2023), it is valid only if the normality approximation is considered valid, which requires sufficient large sample sizes $n_1, n_2$ and dimension $d$. As we discussed before, one rule to use the approximation is when $n_1, n_2 \geq 25, d \geq 50$, as suggested by Gao & Shao (2023).

## 5 Experiments

This section investigates the Type I error and statistical power of MMD-Miss proposed in Section 4, and compare it with other missing data methods, including missForest Stekhoven & Bühlmann (2012), the Wilcoxon-Mann-Whitney test with missing data method Zeng et al. (2024) (WMWM), and three common missing data approaches, namely case deletion, mean imputation and hot deck imputation. All these approaches are described in more detail in Appendix B.3. For MMD-Miss, the parameter $\beta$ in the Laplacian kernel is chosen using the median heuristic, which generally works well (Gretton et al., 2012a; Bodenham & Kawahara, 2023) and is described in Appendix B.2. The number of permutations used for MMD-Perm and the imputation methods is set to $B = 100$, as described in Appendix B.2.

**Asymptotic Type I error for a small proportion of missing data (5%).** The first experiment investigates the Type I error of the proposed method MMD-Miss and the common missing data approaches, as the sample size increases, while keeping the proportion of missing data fixed at 5%. In this experiment, the data are missing not at random (MNAR). To assess the Type I error, random samples $X = \{x_1, \ldots, x_{n_1}\}$, $Y = \{y_1, \ldots, y_{n_2}\}$ are independently generated from a $d$-dimensional normal distribution with mean vector $\mu = (0, \ldots, 0)^T$ and covariance matrix equal to the identity matrix. Subsequently, 5% of data will be selected in both $X$ and $Y$ to be labelled as missing (for $d = 1$), or incomplete (for $d > 1$). For multivariate data, when an observation is labelled as incomplete, 30% of its components will be missing values. We select 5% of data to be missing or incomplete to show that even small proportion of missing data in the data set could lead to invalid testing results, if the missing data are not taken into account. Similarly, for multivariate data, the 30% missingness per incomplete observation is chosen to introduce a moderate amount of missing data.

We run this experiment for different dimensions $d \in \{1, 10, 50\}$. In the univariate case, when the dimension $d = 1$, observations $x_i \in X$ with $x_i < 0$ will be randomly selected to possibly be missing, while only observations $y_i \in Y$ with $y_i > 0$ will be randomly selected to possibly be missing; in this case the data are missing not at random (MNAR), i.e. the data are informatively missing. In the multivariate cases, when $d \in \{10, 50\}$, the observations in $X$ that will possibly be partially missing are those with $\sum_{l=1}^{d} x_i(l)/\sqrt{d} < -0.8$, and then for each chosen sample, 30% of its components with values smaller than $\text{median}(x_i(1), \ldots, x_i(d))$ will be randomly selected to be missing. In other words, only observations with low total values, i.e., with a sum smaller than $-0.8\sqrt{d}$ have a chance to be incomplete, and within these, components that fall below the median will be missing. This design creates MNAR patterns, where the presence of the values depend on both missing and observed data. On the other hand, only observations in $Y$ with $\sum_{l=1}^{d} y_i(l)/\sqrt{d} > 0.8$ will possibly be partially missing, and again for each chosen observation, 30% its components with values larger than $\text{median}(y_i(1), \ldots, y_i(d))$ will be randomly selected to be missing. We use these missingness mechanisms, since if only the fully observed observations are taken into account, the two samples will appear to be different.

Figure 1 shows that MMD-Miss-Perm controls the Type I error for dimension $d \in \{1, 10, 50\}$. MMD-Miss-Normality is only suitable for $d = 50$, and controls the Type I error in this case. WMWM (Zeng et al., 2024) controls the Type I error for $d = 1$, but is not suitable for multivariate data. All imputation methods fail to control the Type I error. Note that MissForest (Stekhoven & Bühlmann, 2012) is only suitable for multivariate data, but similarly to the other imputation methods, it fails to control the Type I error. In particular, Figure 1 shows that even when only 5% of the data are missing, as the sample size increases to 5000, in these experiments the Type I error of all imputation methods approaches 1.

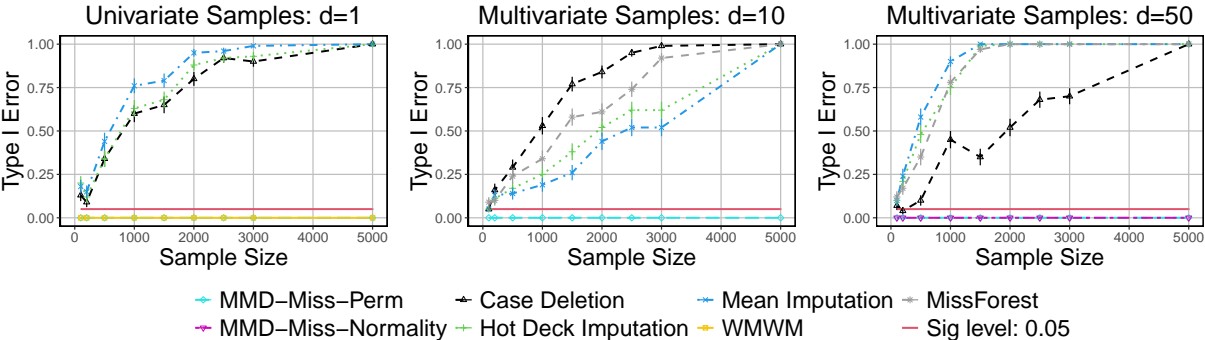

Figure 1: The Type I error of the proposed method and common missing data methods when data are missing not at random. (Left): $d = 1$; (Middle): $d = 10$; (Right): $d = 50$. MMD-Miss-Normality is considered when $d = 50$. For all figures, the significance level is $\alpha = 0.05$, and 5% of the data are missing or partially observed. The displayed values show the average times of the null hypothesis is rejected over 100 repetitions. The error bars represent one standard error of the mean. Note that WMWM is only applicable for univariate data, while MissForest in only applicable for multivariate data.

**Type I error and power for univariate data.** The second experiment investigates the Type I error and statistical power for a fixed sample size, while the proportion of missing data increases. In this experiment, the data are again missing not at random (MNAR). To assess the Type I error, the data $X = \{x_1, \ldots, x_{n_1}\}$ and $Y = \{y_1, \ldots, y_{n_2}\}$ are independently sampled from a standard normal distribution $N(0, 1)$. To assess the statistical power, $X$ are sampled independently from $N(0, 1)$ while $Y$ are sampled independently from either (a) $N(1, 1)$ or (b) $N(1.5, 1)$. Then, a proportion $s \in \{0, 0.01, \ldots, 0.20\}$ of samples are selected in $X$ and $Y$ to be missing. The missingness mechanisms are the same as for Figure 1 when $d = 1$; only observations $x_i \in X$ with $x_i < 0$ and observations $y_i \in Y$ with $y_i > 0$ will be randomly selected to possibly be missing.

Figure 2 shows that the Type I error is not controlled by the case deletion, hot deck imputation or mean imputation methods. When the proportion of missing data is $s = 5\%$, the Type I error is above 25% for each of these three methods, and when the $s = 10\%$ the Type I error of these three methods is beyond 75%. On the other hand, the proposed method MMD-Miss-Perm controls the Type I error, regardless of the proportion of missing data. While all the common missing data methods have good statistical power, the power of the proposed method decreases significantly when there are more than 5% missing data for alternative $N(0, 1)$ vs $N(1, 1)$, and when there are more than 10% missing data for alternative $N(0, 1)$ vs $N(1.5, 1)$. This experiment demonstrates that MMD-Miss-Perm is useful when the proportion of missing data is in the range 5% to 10%, while the three common missing data approaches fail to control the Type I error.

We note that WMWM (Zeng et al., 2024) controls the Type I error in all cases, and has better power than MMD-Miss-Perm when there is a change in the mean. However, it has almost zero power when there is a change in variance (bottom row of Figure 2); this is not surprising, since it is based on the Wilcoxon-Mann-Whitney test, which is designed to test for a change in location. On the other hand, MMD-Miss-Perm controls the Type I error and has good statistical power when 5% to 10% of the data are missing, both when there is a change in mean, and when there is a change in variance.

**Remark 2.** As reflected in these results, our proposed method is guaranteed to control the Type I error, even in cases where data are missing not at random. However, the cost is that our method is conservative, since it only declares a significant result when the maximum possible $p$-value, considered over all possible imputations, is significant. When the proportion of missing data increases, the upper and lower bounds of the MMD test statistic increase and decrease, respectively, and so the power decreases.

**Type I error and power for multivariate data.** The third experiment compares the Type I error and power of MMD-Miss and the three common missing data approaches for multivariate observations. For a value $a \in \mathbb{R}$, let $\mu_{a,d} = (a, a, \ldots, a)^T$ be a mean vector with $d$ components all equal to $a$. To assess the

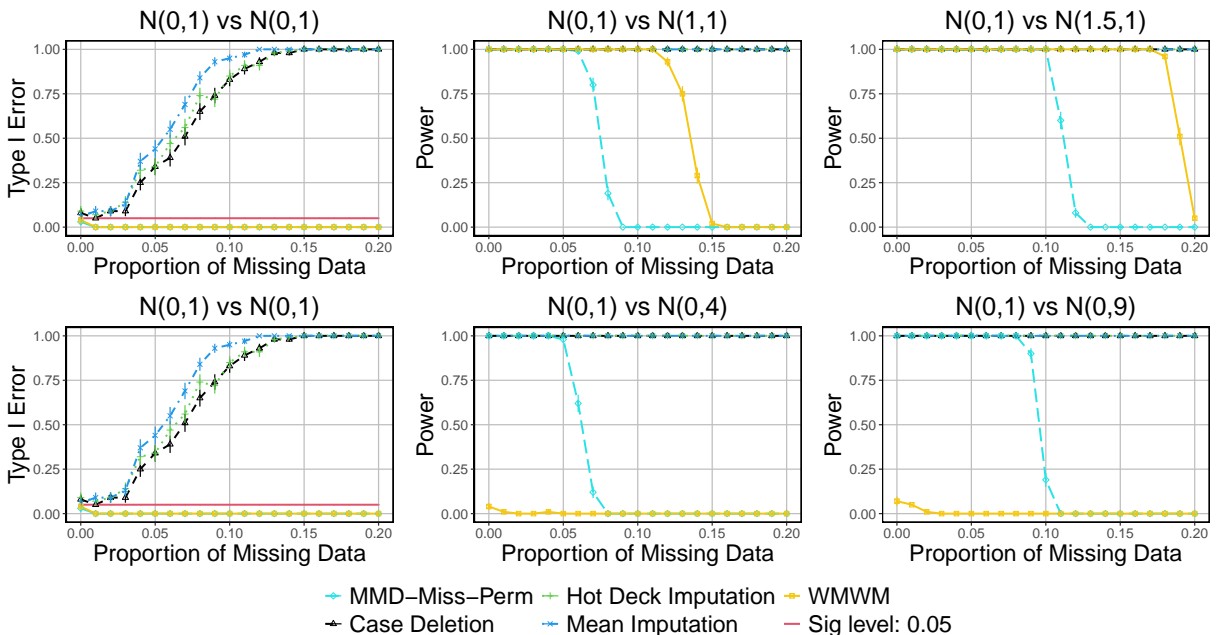

Figure 2: The Type I error and power of the proposed method and common missing data methods for univariate samples when data are missing not at random. (Left): N(0, 1) vs N(0, 1); (Middle): N(0, 1) vs N(1, 1); (Right): N(0, 1) vs N(1.5, 1). A significance level $\alpha = 0.05$ and sample sizes $n_1 = n_2 = 500$ are used. The displayed values show the average times of the null hypothesis is rejected over 100 repetitions. The error bars represent one standard error of the mean. The Type I error figure in the left-hand column appears in each row for convenience of comparison.

Type I error, observations $X = \{x_1, \ldots, x_{n_1}\}$ and $Y = \{y_1, \ldots, y_{n_2}\}$ are independently sampled from the $d$-dimensional normal distribution with zero mean vector $\mu_{0,d} = (0, 0, \ldots, 0)^T$ and covariance matrix $\Sigma = I_d$, the identity matrix. To assess the statistical power, $X$ are independently sampled from $N(\mu_{0,d}, I_d)$ and $Y$ are samples independently from either (a) $N(\mu_{1,d}, I_d)$ or (b) $N(\mu_{1.5,d}, I_d)$. Then, a proportion $s \in [0, 0.2]$ of observations from both $X$ and $Y$ will be randomly selected to be partially missing. The missingness mechanisms follows the same approach used for Figure 1 when $d > 1$. The results displayed in Figure 3 show a similar pattern to the results for univariate case: (i) the three common missing data approaches cannot control the Type I error although they all demonstrate good power, while (ii) MMD-Miss controls the Type I error, and also have good power when proportion of missing data is around 5% to 10%. Notably, in the case where $d = 50$, MMD-Miss-Normality has better statistical power than MMD-Miss-Perm. For the alternative $N((1.5, \ldots, 1.5)^T, I_d)$, MMD-Miss-Normality approximation has good power when 15% of the data have missing values.

**MNIST dataset.** We evaluate the performance of MMD-Miss on real-world data using MNIST images LeCun et al. (1998), with examples shown in Figure 4 in Appendix B.4. Each image in the MNIST dataset has dimensions of $28 \times 28$ pixels and is labelled from 0 to 9. For our analysis, the pixel values of each image are scaled between 0 and 1. To assess Type I error, datasets $X$ and $Y$ are generated by randomly sampling with replacement from MNIST training set images labelled as 3. For evaluating power, $X$ is generated by randomly sampling with replacement from images labelled as 0, while $Y$ continues to be sampled from images labelled as 3. A proportion $s \in [0, 0.25]$ of the samples in $Y$ are then randomly selected and labelled as incomplete if there are more than 85 non-zero pixels in the region defined by rows 1 to 14 and columns 8 to 21 (i.e. a sub-block in the upper half of each image), which will be marked as missing. In other words, images with more non-zero pixels in the specified region are more likely to be partially observed. Examples of these incompletely observed images are shown in Figure 5 in Appendix B.4 Table 1 shows that the three common

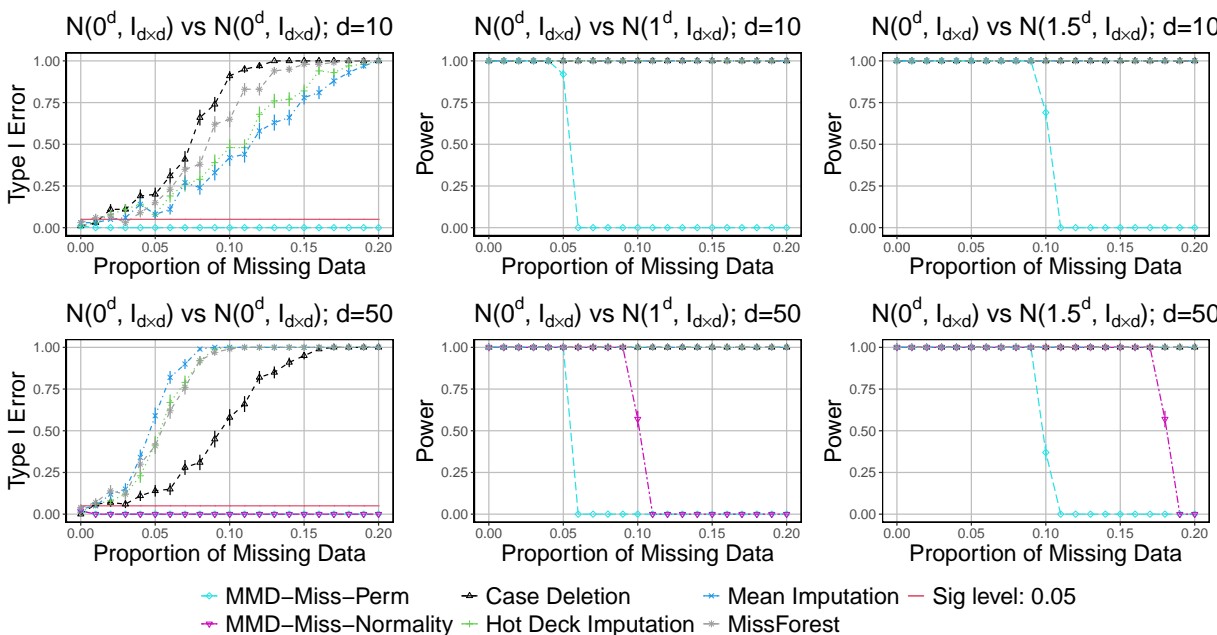

Figure 3: The Type I error and power of MMD-Miss and the three common missing data approaches for multivariate samples when data are missing not at random. The dimension of the data is either $d = 10$ or $d = 50$. When $d = 50$, the MMD-Miss-Normality is used. A significance level $\alpha = 0.05$, and sample sizes $n_1 = n_2 = 500$ are used. The displayed values show the average times of the null hypothesis is rejected over 100 repetitions. The error bars represent one standard error of the mean.

Table 1: Comparison of Type I error and power on MNIST dataset. NA stands for MMD-Miss-Normality; PT for MMD-Miss-Perm; CD stands for case deletion; MI stands for mean imputation; HD stands for hot deck imputation. A significance level $\alpha = 0.05$, and sample sizes $n_1 = n_2 = 500$ are used. The values in the table are the average times the null hypothesis is rejected over 100 repetitions.

| Proportion of Missing Data ($s$) | Type I Error | | | | | Power | | | | |
| | MMD-Miss | | Standard | | | MMD-Miss | | Standard | | |
| | NA | PT | CD | MI | HD | NA | PT | CD | MI | HD |
|---|---|---|---|---|---|---|---|---|---|---|
| 0.00 | 0.07 | 0.07 | 0.06 | 0.07 | 0.07 | 1.00 | 1.00 | 1.00 | 1.00 | 1.00 |
| 0.05 | 0.00 | 0.00 | 0.08 | 1.00 | 1.00 | 1.00 | 1.00 | 1.00 | 1.00 | 1.00 |
| 0.10 | 0.00 | 0.00 | 0.21 | 1.00 | 1.00 | 1.00 | 1.00 | 1.00 | 1.00 | 1.00 |
| 0.15 | 0.00 | 0.00 | 0.64 | 1.00 | 1.00 | 1.00 | 0.00 | 1.00 | 1.00 | 1.00 |
| 0.20 | 0.00 | 0.00 | 0.91 | 1.00 | 1.00 | 0.96 | 0.00 | 1.00 | 1.00 | 1.00 |
| 0.25 | 0.00 | 0.00 | 1.00 | 1.00 | 1.00 | 0.00 | 0.00 | 1.00 | 1.00 | 1.00 |

missing data approaches cannot control the Type I error for this task. On the other hand, MMD-Miss controls the Type I error while enjoying good power. For MMD-Miss-Normality, the power is good except for the case when 25% images from $Y$ are missing.

## 6 Conclusion

The proposed method MMD-Miss performs two-sample testing on both univariate and multivariate data with missing values for any distributional shifts. This method is based on deriving precise bounds of the Laplacian

kernel, and rejects the null hypothesis when all possible missing data lead to significant results. MMD-Miss controls the Type I error, while other common missing data methods such as case deletion fail to control the Type I error when the data are missing not at random (MNAR). One limitation of MMD-Miss is that it is restricted to the Laplacian kernel for computational reasons. However, the Laplacian kernel is characteristic, and MMD with this kernel can detect any distributional change. Moreover, this approach will be effective for other kernels if appropriate bounds can be derived and computed efficiently.

Simulation results show that MMD-Miss has good statistical power, typically when 5% to 10% of the data are missing. Our experiments show that in high-dimensional situations, i.e. $d \geq 50$, using a normal approximation (Gao & Shao, 2023) (MMD-Normality) improves the statistical power of MMD-Miss based on permutation test (MMD-Perm). Another limitation of MMD-Miss is that it is only suitable when up to 20% of the data is missing. However, our experiments show that using imputation methods when even 5% of the data are missing not at random can lead to the Type I error not being controlled. Overall, MMD-Miss is a robust approach to two-sample testing in the presence of missing data for detecting changes in univariate or multivariate data

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

## Overview of Appendix

Appendix A. Mathematical details of results given in Section 3 and Section 4.

Appendix B. Details of experiments in Section 5.

Appendix C. Additional experiments.

## A    Mathematical Details

### A.1    Decomposition of MMD

Without loss of generality, let us assume $x_1, \ldots, x_{m_1}, y_1, \ldots, y_{m_2}$ are data that are not observed, or not fully observed. Then, $\mathrm{MMD}_u^2(X, Y)$ can be decomposed, using the following lemma:

**Lemma 5.** *Suppose $X = \{x_1, \ldots, x_{n_1}\}$, $Y = \{y_1, \ldots, y_{n_2}\}$ are subsets of $\mathbb{R}^d$, where $d \geq 1$. Suppose that $MMD_u^2(X, Y)$ is unbiased MMD test statistic defined as*

$$MMD_u^2(X, Y) = \tfrac{1}{n_1(n_1-1)} \sum_{i=1}^{n_1} \sum_{\substack{j=1 \\ j \neq i}}^{n_2} k(x_i, x_j) + \tfrac{1}{n_2(n_2-1)} \sum_{i=1}^{n_2} \sum_{\substack{j=1 \\ j \neq i}}^{n_2} k(y_i, y_j) - \tfrac{2}{n_1 n_2} \sum_{i=1}^{n_1} \sum_{j=1}^{n_2} k(x_i, y_j),$$

*with $k$ denoting the Laplacian kernel. Then, for any two positive integers $m_1$, $m_2$ such that $m_1 \leq n_1$, $m_2 \leq n_2$, $MMD_u^2(X, Y)$ can be rewritten as:*

$$MMD_u^2(X, Y) = A_1 + A_2 + A_3 + A_4,$$

*where $c_1 = \tfrac{2}{n_1(n_1-1)}$, $c_2 = \tfrac{2}{n_2(n_2-1)}$, $c_3 = \tfrac{2}{n_1 n_2}$, and*

$$A_1 = c_1 \sum_{i=1}^{m_1} \sum_{j=i+1}^{m_1} k(x_i, x_j) + c_2 \sum_{i=1}^{m_2} \sum_{j=i+1}^{m_2} k(y_i, y_j) - c_3 \sum_{i=1}^{m_1} \sum_{j=1}^{m_2} k(x_i, y_j),$$

$$A_2 = c_1 \sum_{i=m_1+1}^{n_1-1} \sum_{j=i+1}^{n_1} k(x_i, x_j) + c_2 \sum_{i=m_2+1}^{n_2-1} \sum_{j=i+1}^{n_2} k(y_i, y_j) - c_3 \sum_{i=m_1+1}^{n_1} \sum_{j=m_2+1}^{n_2} k(x_i, y_j),$$

$$A_3 = c_1 \sum_{i=1}^{m_1} \sum_{j=m_1+1}^{n_1} k(x_i, x_j) - c_3 \sum_{i=1}^{m_1} \sum_{j=m_2+1}^{n_2} k(x_i, y_j),$$

$$A_4 = c_2 \sum_{i=1}^{m_2} \sum_{j=m_2+1}^{n_2} k(y_i, y_j) - c_3 \sum_{i=m_1+1}^{n_1} \sum_{j=1}^{m_2} k(x_i, y_j).$$

*Proof.* To start, for any $x$, $y$ of $\mathbb{R}^d$ real values, the Laplacian kernel is defined as follows

$$k(x, y) = \exp(-\beta ||x - y||_1).$$

It can be seen that $k$ is symmetric, i.e. $k(x, y) = k(y, x)$. Subsequently, according to Lemma 2 in Bodenham & Kawahara (2023), the $\mathrm{MMD}_u^2(X, Y)$ can then be rewritten as

$$\mathrm{MMD}_u^2(X, Y) = \frac{2}{n_1(n_1-1)} \sum_{i=1}^{n_1} \sum_{j=1}^{i-1} k(x_i, x_j) + \frac{2}{n_2(n_2-1)} \sum_{i=1}^{n_2} \sum_{j=1}^{i-1} k(y_i, y_j)$$

$$- \frac{2}{n_1 n_2} \sum_{i=1}^{n_1} \sum_{j=1}^{n_2} k(x_i, y_j).$$

Notice that

$$\sum_{i=1}^{n_1}\sum_{j=1}^{i-1} k(x_i, x_j) = \sum_{i=1}^{n_1-1}\sum_{j=i+1}^{n_1} k(x_i, x_j),$$

$$\sum_{i=1}^{n_2}\sum_{j=1}^{i-1} k(y_i, y_j) = \sum_{i=1}^{n_2-1}\sum_{j=i+1}^{n_2} k(y_i, y_j).$$

Thus, we further have

$$\text{MMD}_u^2(X, Y) = \frac{2}{n_1(n_1-1)}\sum_{i=1}^{n_1-1}\sum_{j=i+1}^{n_1} k(x_i, x_j) + \frac{2}{n_2(n_2-1)}\sum_{i=1}^{n_2-1}\sum_{j=i+1}^{n_2} k(y_i, y_j)$$

$$- \frac{2}{n_1 n_2}\sum_{i=1}^{n_1}\sum_{j=1}^{n_2} k(x_i, y_j).$$

Let $m_1, m_2$ be any two positive integers such that $m_1 \leq n_1, m_2 \leq n_2$. Denote $c_1 = \frac{2}{n_1(n_1-1)}$. Subsequently,

$$\frac{2}{n_1(n_1-1)}\sum_{i=1}^{n_1-1}\sum_{j=i+1}^{n_1} k(x_i, x_j) = c_1\sum_{i=1}^{m_1}\sum_{j=i+1}^{n_1} k(x_i, x_j) + c_1\sum_{i=m_1+1}^{n_1-1}\sum_{j=i+1}^{n_1} k(x_i, x_j)$$

$$= c_1\sum_{i=1}^{m_1}\sum_{j=i+1}^{m_1} k(x_i, x_j) + c_1\sum_{i=1}^{m_1}\sum_{j=m_1+1}^{n_1} k(x_i, x_j)$$

$$+ c_1\sum_{i=m_1+1}^{n_1-1}\sum_{j=i+1}^{n_1} k(x_i, x_j).$$

Also, denote $c_2 = \frac{2}{n_2(n_2-1)}$, it follows that

$$\frac{2}{n_2(n_2-1)}\sum_{i=1}^{n_2-1}\sum_{j=i+1}^{n_2} k(y_i, y_j) = c_2\sum_{i=1}^{m_2}\sum_{j=i+1}^{n_2} k(y_i, y_j) + c_2\sum_{i=m_2+1}^{n_2-1}\sum_{j=i+1}^{n_2} k(y_i, y_j)$$

$$= c_2\sum_{i=1}^{m_2}\sum_{j=i+1}^{m_2} k(y_i, y_j) + c_2\sum_{i=1}^{m_2}\sum_{j=m_2+1}^{n_2} k(y_i, y_j)$$

$$+ c_2\sum_{i=m_2+1}^{n_2-1}\sum_{j=i+1}^{n_2} k(y_i, y_j).$$

Denote $c_3 = \frac{2}{n_1 n_2}$, then

$$\frac{2}{n_1 n_2}\sum_{i=1}^{n_1}\sum_{j=1}^{n_2} k(x_i, y_j) = c_3\sum_{i=1}^{m_1}\sum_{j=1}^{n_2} k(x_i, y_j) + c_3\sum_{i=m_1+1}^{n_1}\sum_{j=1}^{n_2} k(x_i, y_j)$$

$$= c_3\sum_{i=1}^{m_1}\sum_{j=1}^{m_2} k(x_i, y_j) + c_3\sum_{i=1}^{m_1}\sum_{j=m_2+1}^{n_2} k(x_i, y_j)$$

$$+ c_3\sum_{i=m_1+1}^{n_1}\sum_{j=1}^{m_2} k(x_i, y_j) + c_3\sum_{i=m_1+1}^{n_1}\sum_{j=m_2+1}^{n_2} k(x_i, y_j).$$

Combine the above together, $\text{MMD}_u^2(X, Y)$ can be rewritten as

$$
\begin{aligned}
\text{MMD}_u^2(X, Y) = {} & c_1 \sum_{i=1}^{m_1} \sum_{j=i+1}^{m_1} k(x_i, x_j) + c_1 \sum_{i=1}^{m_1} \sum_{j=m_1+1}^{n_1} k(x_i, x_j) \\
& + c_1 \sum_{i=m_1+1}^{n_1-1} \sum_{j=i+1}^{n_1} k(x_i, x_j) + c_2 \sum_{i=1}^{m_2} \sum_{j=i+1}^{m_2} k(y_i, y_j) \\
& + c_2 \sum_{i=1}^{m_2} \sum_{j=m_2+1}^{n_2} k(y_i, y_j) + c_2 \sum_{i=m_2+1}^{n_2-1} \sum_{j=i+1}^{n_2} k(y_i, y_j) \\
& - c_3 \sum_{i=1}^{m_1} \sum_{j=1}^{m_2} k(x_i, y_j) - c_3 \sum_{i=1}^{m_1} \sum_{j=m_2+1}^{n_2} k(x_i, y_j) \\
& - c_3 \sum_{i=m_1+1}^{n_1} \sum_{j=1}^{m_2} k(x_i, y_j) - c_3 \sum_{i=m_1+1}^{n_1} \sum_{j=m_2+1}^{n_2} k(x_i, y_j).
\end{aligned}
$$

By rearranging the above equation, we conclude that

$$
\begin{aligned}
\text{MMD}_u^2(X, Y) = {} & c_1 \sum_{i=1}^{m_1} \sum_{j=i+1}^{m_1} k(x_i, x_j) + c_2 \sum_{i=1}^{m_2} \sum_{j=i+1}^{m_2} k(y_i, y_j) - c_3 \sum_{i=1}^{m_1} \sum_{j=1}^{m_2} k(x_i, y_j) \\
& + c_1 \sum_{i=m_1+1}^{n_1-1} \sum_{j=i+1}^{n_1} k(x_i, x_j) + c_2 \sum_{i=m_2+1}^{n_2-1} \sum_{j=i+1}^{n_2} k(y_i, y_j) - c_3 \sum_{i=m_1+1}^{n_1} \sum_{j=m_2+1}^{n_2} k(x_i, y_j) \\
& + c_1 \sum_{i=1}^{m_1} \sum_{j=m_1+1}^{n_1} k(x_i, x_j) - c_3 \sum_{i=1}^{m_1} \sum_{j=m_2+1}^{n_2} k(x_i, y_j) \\
& - c_2 \sum_{i=1}^{m_2} \sum_{j=m_2+1}^{n_2} k(y_i, y_j) - c_3 \sum_{i=m_1+1}^{n_1} \sum_{j=1}^{m_2} k(x_i, y_j),
\end{aligned}
$$

which proves our result. $\qquad\square$

## A.2  Proof of Lemma 2

**Lemma 6.** *Let $x_1, \ldots, x_{\ell_1}$ and $y_1, \ldots, y_{\ell_2}$ be univariate real values, that are observed. Suppose $a_1, \ldots, a_{\ell_1}$, $b_1, \ldots, b_{\ell_2}$ and $\beta$ are positive constants. Define*

$$
T(z) = \sum_{i=1}^{\ell_1} a_i \exp(-\beta|x_i - z|) - \sum_{i=1}^{\ell_2} b_i \exp(-\beta|y_i - z|)
$$

*as a function of $z \in \mathbb{R}$. Subsequently, for any given $z_0 \in \mathbb{R}$,*

$$
T(z_0) \geq \min\{0, T(x_1), \ldots, T(x_{\ell_1}), T(y_1), \ldots, T(y_{\ell_2})\}. \tag{9}
$$

*On the other hand,*

$$
T(z_0) \leq \max\{0, T(x_1), \ldots, T(x_{\ell_1}), T(y_1), \ldots, T(y_{\ell_2})\}. \tag{10}
$$

*Proof.* We will first prove that inequality 9 holds. Let $z_1 = \min\{x_1, \ldots, x_{\ell_1}, y_1, \ldots, y_{\ell_2}\}$. When $z < z_1$, it follows

$$
\begin{aligned}
T(z) &= \sum_{i=1}^{\ell_1} a_i \exp(-\beta(x_i - z)) - \sum_{i=1}^{\ell_2} b_i \exp(-\beta(y_i - z)) \\
&= \sum_{i=1}^{\ell_1} a_i \exp(-\beta(x_i - z_1 + z_1 - z)) - \sum_{i=1}^{\ell_2} b_i \exp(-\beta(y_i - z_1 + z_1 - z)) \\
&= \sum_{i=1}^{\ell_1} a_i \exp(-\beta(x_i - z_1)) \exp(-\beta(z_1 - z)) - \sum_{i=1}^{\ell_2} b_i \exp(-\beta(y_i - z_1)) \exp(-\beta(z_1 - z)) \\
&= \exp(-\beta(z_1 - z)) \left\{ \sum_{i=1}^{\ell_1} a_i \exp(-\beta(x_i - z_1)) - \sum_{i=1}^{\ell_2} b_i \exp(-\beta(y_i - z_1)) \right\} \\
&= \exp(-\beta(z_1 - z)) T(z_1).
\end{aligned}
$$

Notice that, since $\beta > 0$ and $z < z_1$,

$$
0 < \exp(-\beta(z_1 - z)) < 1.
$$

Hence, if $T(z_1) \geq 0$,

$$
\begin{aligned}
0 \leq \exp(-\beta(z_1 - z)) T(z_1) \leq T(z_1); \\
\Longrightarrow 0 \leq T(z)
\end{aligned}
$$

if $T(z_1) < 0$,

$$
\begin{aligned}
0 > \exp(-\beta(z_1 - z)) T(z_1) > T(z_1); \\
\Longrightarrow T(z) > T(z_1).
\end{aligned}
$$

Thus, for any given $z_0 \in \mathbb{R}$, if $z_0 \in (-\infty, z_1)$, we have

$$
T(z_0) \geq \min\{0, T(z_1)\},
$$

which proves equation 9 when $z_0 \in (-\infty, z_1)$.

Similarly, let $z_2 = \max\{x_1, \ldots, x_{\ell_1}, y_1, \ldots, y_{\ell_2}\}$. When $z > z_2$, it follows

$$
\begin{aligned}
T(z) &= \sum_{i=1}^{\ell_1} a_i \exp(-\beta(z - x_i)) - \sum_{i=1}^{\ell_2} b_i \exp(-\beta(z - y_i)) \\
&= \sum_{i=1}^{\ell_1} a_i \exp(-\beta(z - z_2 + z_2 - x_i)) - \sum_{i=1}^{\ell_2} b_i \exp(-\beta(z - z_2 + z_2 - y_i)) \\
&= \sum_{i=1}^{\ell_1} a_i \exp(-\beta(z_2 - x_i)) \exp(-\beta(z - z_2)) - \sum_{i=1}^{\ell_2} b_i \exp(-\beta(z_2 - y_i)) \exp(-\beta(z - z_2)) \\
&= \exp(-\beta(z - z_2)) \left\{ \sum_{i=1}^{\ell_1} a_i \exp(-\beta(z_2 - x_i)) - \sum_{i=1}^{\ell_2} b_i \exp(-\beta(z_2 - y_i)) \right\} \\
&= \exp(-\beta(z - z_2)) T(z_2).
\end{aligned}
$$

Notice that, since $\beta > 0$ and $z > z_2$,

$$
0 < \exp(-\beta(z - z_2)) < 1.
$$

Hence, if $T(z_2) \geq 0$,

$$0 \leq \exp(-\beta(z - z_2))T(z_2) \leq T(z_2);$$
$$\implies 0 \leq T(z)$$

if $T(z_1) < 0$,

$$0 > \exp(-\beta(z - z_2))T(z_2) > T(z_2);$$
$$\implies T(z) > T(z_2).$$

Thus, for any given $z_0 \in \mathbb{R}$, if $z_0 \in (z_2, \infty)$, we have

$$T(z_0) > \min\{0, T(z_2)\},$$

which proves equation 9 when $z_0 \in (z_2, \infty)$.

Suppose $z_1 < z_0 < z_2$ and $z_0 \notin \{x_1, \ldots, x_{\ell_1}, y_1, \ldots, y_{\ell_2}\}$. Then, there must be at least one number in $\{x_1, \ldots, x_{\ell_1}, y_1, \ldots, y_{\ell_2}\}$ smaller than $z$, and at least one number larger than $z$. Subsequently, denote $z_3$ as the maximum number in $\{x_1, \ldots, x_{\ell_1}, y_1, \ldots, y_{\ell_2}\}$ smaller than $z_0$; denote $z_4$ as the minimum number in $\{x_1, \ldots, x_{\ell_1}, y_1, \ldots, y_{\ell_2}\}$ larger than $z_0$.

Notice that

$$T(z) = \sum_{i=1}^{\ell_1} a_i \exp(-\beta|x_i - z|) - \sum_{i=1}^{\ell_2} b_i \exp(-\beta|y_i - z|)$$
$$= \sum_{i=1}^{\ell_1} a_i \mathbb{I}(x_i \leq z_3) \exp(-\beta|x_i - z|) + \sum_{i=1}^{\ell_1} a_i \mathbb{I}(x_i > z_3) \exp(-\beta|x_i - z|)$$
$$- \sum_{i=1}^{\ell_2} b_i \mathbb{I}(y_i \leq z_3) \exp(-\beta|y_i - z|) - \sum_{i=1}^{\ell_2} b_i \mathbb{I}(y_i > z_3) \exp(-\beta|y_i - z|).$$

When $z_3 < z < z_4$, according to the definition of $z_3$ and $z_4$, for any $x_i, y_i \leq z_3$, it follows $z > x_i, y_i$. On the other hand, for for any $x_i, y_i > z_3$, it follows $x_i, y_i \geq z_4$, which gives $z < x_i, y_i$.

Hence, when $z_3 < z < z_4$,

$$T(z) = \sum_{i=1}^{\ell_1} a_i \mathbb{I}(x_i \leq z_3) \exp(-\beta(z - x_i)) + \sum_{i=1}^{\ell_1} a_i \mathbb{I}(x_i > z_3) \exp(-\beta(x_i - z))$$
$$- \sum_{i=1}^{\ell_2} b_i \mathbb{I}(y_i \leq z_3) \exp(-\beta(z - y_i)) - \sum_{i=1}^{\ell_2} b_i \mathbb{I}(y_i > z_3) \exp(-\beta(y_i - z)).$$

Further,

$$T(z) = \sum_{i=1}^{\ell_1} a_i \mathbb{I}(x_i \le z_3) \exp(-\beta(z - z_3 + z_3 - x_i)) + \sum_{i=1}^{\ell_1} a_i \mathbb{I}(x_i > z_3) \exp(-\beta(x_i - z_4 + z_4 - z))$$

$$- \sum_{i=1}^{\ell_2} b_i \mathbb{I}(y_i \le z_3) \exp(-\beta(z - z_3 + z_3 - y_i)) - \sum_{i=1}^{\ell_2} b_i \mathbb{I}(y_i > z_3) \exp(-\beta(y_i - z_4 + z_4 - z)).$$

$$= \exp(-\beta(z - z_3)) \sum_{i=1}^{\ell_1} a_i \mathbb{I}(x_i \le z_3) \exp(-\beta(z_3 - x_i))$$

$$+ \exp(-\beta(z_4 - z)) \sum_{i=1}^{\ell_1} a_i \mathbb{I}(x_i > z_3) \exp(-\beta(x_i - z_4))$$

$$- \exp(-\beta(z - z_3)) \sum_{i=1}^{\ell_2} b_i \mathbb{I}(y_i \le z_3) \exp(-\beta(z_3 - y_i))$$

$$- \exp(-\beta(z_4 - z)) \sum_{i=1}^{\ell_2} b_i \mathbb{I}(y_i > z_3) \exp(-\beta(y_i - z_4)).$$

$$= \exp(-\beta(z - z_3)) \left\{ \sum_{i=1}^{\ell_1} a_i \mathbb{I}(x_i \le z_3) \exp(-\beta(z_3 - x_i)) - \sum_{i=1}^{\ell_2} b_i \mathbb{I}(y_i \le z_3) \exp(-\beta(z_3 - y_i)) \right\}$$

$$+ \exp(-\beta(z_4 - z)) \left\{ \sum_{i=1}^{\ell_1} a_i \mathbb{I}(x_i > z_3) \exp(-\beta(x_i - z_4)) - \sum_{i=1}^{\ell_2} b_i \mathbb{I}(y_i > z_3) \exp(-\beta(y_i - z_4)) \right\}.$$

For notation ease, let us denote

$$A := \left\{ \sum_{i=1}^{\ell_1} a_i \mathbb{I}(x_i \le z_3) \exp(-\beta(z_3 - x_i)) - \sum_{i=1}^{\ell_2} b_i \mathbb{I}(y_i \le z_3) \exp(-\beta(z_3 - y_i)) \right\},$$

$$B := \left\{ \sum_{i=1}^{\ell_1} a_i \mathbb{I}(x_i > z_3) \exp(-\beta(x_i - z_4)) - \sum_{i=1}^{\ell_2} b_i \mathbb{I}(y_i > z_3) \exp(-\beta(y_i - z_4)) \right\}.$$

Thus,

$$T(z) = A \exp(-\beta(z - z_3)) + B \exp(-\beta(z_4 - z)).$$

Notice that if $B = 0$, for any $z_3 < z < z_4$, $T(z) = A \exp(-\beta(z - z_3))$ is a monotonic increasing function in $(z_3, z_4)$ when $A < 0$, or a monotonic decreasing function in $(z_3, z_4)$ when $A > 0$, or a constant function in $(z_3, z_4)$ when $A = 0$. Thus, we always have $\min\{T(z_3), T(z_4)\} \le T(z_0)$ for $z_0 \in (z_3, z_4)$, which proves inequality 9 directly.

If, however, $B \ne 0$, let $T(z) = 0$, we have

$$A \exp(-\beta(z - z_3)) = -B \exp(-\beta(z_4 - z))$$

$$\implies -\frac{A}{B} = \frac{\exp(-\beta(z_4 - z))}{\exp(-\beta(z - z_3))}.$$

When $-\frac{A}{B} \le 0$, since $\frac{\exp(-\beta(z_4-z))}{\exp(-\beta(z-z_3))} > 0$, then $T(z)$ cannot take 0 in $(z_3, z_4)$. If however, $-\frac{A}{B} > 0$, we further have

$$\log\left(-\frac{A}{B}\right) = -\beta(z_4 - z) - \{-\beta(z - z_3)\}$$

$$\implies \log\left(-\frac{A}{B}\right) = -\beta z_4 + \beta z + \beta z - \beta z_3$$

$$\implies z = \frac{1}{\beta} \log\left(-\frac{A}{B}\right) + z_4 + z_3.$$

That is, $T(z) = 0$ when $z = \frac{1}{\beta} \log\left(-\frac{A}{B}\right) + z_4 + z_3 \in (z_3, z_4)$. Overall, we have shown $T(z)$ takes $0$ at most once in $(z_3, z_4)$ when $B \neq 0$. This result will be used subsequently for proving our final conclusion.

Now, taking derivative of $T(z)$ of $z$,

$$\frac{\partial T(z)}{\partial z} = -\beta A \exp(-\beta(z - z_3)) + \beta B \exp(-\beta(z_4 - z)).$$

Further, taking derivative of $\frac{\partial T(z)}{\partial z}$ of $z$,

$$\frac{\partial^2 T(z)}{\partial^2 z} = \beta^2 A \exp(-\beta(z - z_3)) + \beta^2 B \exp(-\beta(z_4 - z))$$
$$= \beta^2 T(z).$$

We are going to prove $T(z_0) \geq \min\{0, T(z_3), T(z_4)\}$ when $B \neq 0$ using contradiction. Let us assume that

$$T(z_0) < \min\{0, T(z_3), T(z_4)\}. \tag{11}$$

Then, since $\beta > 0$, it must have

$$\left.\frac{\partial^2 T(z)}{\partial^2 z}\right|_{z_0} = \beta^2 T(z_0) < 0.$$

For $\left.\frac{\partial T(z)}{\partial z}\right|_{z_0}$, it is either $(i): \left.\frac{\partial T(z)}{\partial z}\right|_{z_0} < 0$ or $(ii): \left.\frac{\partial T(z)}{\partial z}\right|_{z_0} \geq 0$. We are going to discuss the two cases separately.

Suppose $(i): \left.\frac{\partial T(z)}{\partial z}\right|_{z_0} < 0$. If for any $z \in [z_0, z_4)$, $\frac{\partial^2 T(z)}{\partial^2 z} < 0$. Then, $\frac{\partial T(z)}{\partial z}$ is a monotonic decreasing function of $z$ in $[z_0, z_4)$. Hence, for any $z \in [z_0, z_4)$,

$$\frac{\partial T(z)}{\partial z} < \left.\frac{\partial T(z)}{\partial z}\right|_{z_0} < 0$$

Thus, $T(z)$ is a monotonic decreasing function in $z \in [z_0, z_4)$, giving us

$$T(z_0) > T(z_4),$$

which is contradicted to the assumption equation 11.

If, however, for any $z \in [z_0, z_4)$, $\frac{\partial^2 T(z)}{\partial^2 z} < 0$ does not holds, recall that we have shown $T(z)$ take $0$ at most once in $(z_3, z_4)$ when $B \neq 0$. This then implies there is only one point $t \in [z_0, z_4)$ such that $\left.\frac{\partial^2 T(z)}{\partial^2 z}\right|_t = 0$. Since $\frac{\partial^2 T(z)}{\partial^2 z}$ is a continuous function and $\left.\frac{\partial^2 T(z)}{\partial^2 z}\right|_{z_0} < 0$, we have

$$\frac{\partial^2 T(z)}{\partial^2 z} < 0, \ z \in [z_0, t).$$

Thus, $\frac{\partial T(z)}{\partial z}$ is a monotonic decreasing function in $z \in [z_0, t)$, giving us

$$\frac{\partial T(z)}{\partial z} < \left.\frac{\partial T(z)}{\partial z}\right|_{z_0} < 0, \ z \in [z_0, t). \tag{12}$$

However, notice that we also have

$$\left.\frac{\partial^2 T(z)}{\partial^2 z}\right|_t = 0 > \left.\frac{\partial^2 T(z)}{\partial^2 z}\right|_{z_0},$$
$$\implies T(t) = 0 > T(z_0)$$
$$\implies \exists t' \in (z_0, t) \text{ such that } \left.\frac{\partial T(z)}{\partial z}\right|_{t'} = \frac{T(t) - T(z_0)}{t - z_0} > 0, \quad \text{(mean value theorem)}$$

which contradicts equation 12.

Suppose $(ii)$ : $\left.\frac{\partial T(z)}{\partial z}\right|_{z_0} \geq 0$. The prove method is similar as to when $(i)$ holds.

If for any $z \in (z_3, z_0]$, $\frac{\partial^2 T(z)}{\partial^2 z} < 0$. Then, $\frac{\partial T(z)}{\partial z}$ is a monotonic decreasing function of $z$ in $(z_3, z_0]$. Hence, for any $z \in (z_3, z_0]$,

$$\frac{\partial T(z)}{\partial z} > \left.\frac{\partial T(z)}{\partial z}\right|_{z_0} \geq 0.$$

Thus, $T(z)$ is a monotonic increasing function in $z \in (z_3, z_0]$, giving

$$T(z_3) < T(z_0),$$

which contradicts to the assumption equation 11.

If, however, for any $z \in (z_3, z_0]$, $\frac{\partial^2 T(z)}{\partial^2 z} < 0$ does not holds, recall that we have shown $T(z)$ taken 0 at most once in $(z_3, z_4)$. This then implies there is only one point $t \in (z_3, z_0]$ such that $\left.\frac{\partial^2 T(z)}{\partial^2 z}\right|_t = 0$. Since $\frac{\partial^2 T(z)}{\partial^2 z}$ is a continuous function and $\left.\frac{\partial^2 T(z)}{\partial^2 z}\right|_{z_0} < 0$, we have

$$\frac{\partial^2 T(z)}{\partial^2 z} < 0, \ z \in (t, z_0].$$

Thus, $\frac{\partial T(z)}{\partial z}$ is a monotonic decreasing function in $z \in (t, z_0]$, giving us

$$\frac{\partial T(z)}{\partial z} > \left.\frac{\partial T(z)}{\partial z}\right|_{z_0} \geq 0, \ z \in (t, z_0]. \tag{13}$$

However, notice that we also have

$$\left.\frac{\partial^2 T(z)}{\partial^2 z}\right|_t = 0 > \left.\frac{\partial^2 T(z)}{\partial^2 z}\right|_{z_0},$$

$$\implies T(t) = 0 > T(z_0)$$

$$\implies \exists t' \in (t, z_0) \text{ such that } \left.\frac{\partial T(z)}{\partial z}\right|_{t'} = \frac{T(t) - T(z_0)}{t - z_0} < 0. \quad \text{(mean value theorem)}$$

which contradicts 13. Hence, we finish our prove for inequality 9.

We now prove inequality 10. Notice that

$$-T(z) = \sum_{i=1}^{\ell_2} b_i \exp(-\beta|y_i - z|) - \sum_{i=1}^{\ell_1} a_i \exp(-\beta|x_i - z|).$$

Subsequently, using the result of inequality 9, we have

$$-T(z_0) \geq \min\{0, -T(x_1), \ldots, -T(x_{\ell_1}), -T(y_1), \ldots, -T(y_{\ell_2})\}$$
$$\implies T(z_0) \leq -\min\{0, -T(x_1), \ldots, -T(x_{\ell_1}), -T(y_1), \ldots, -T(y_{\ell_2})\}$$
$$\implies T(z_0) \leq \max\{0, T(x_1), \ldots, T(x_{\ell_1}), T(y_1), \ldots, T(y_{\ell_2})\},$$

which proves inequality 10. $\qquad\square$

### A.3 Proof of Lemma 3

This section proves Lemma 3, extending Lemma 2 into higher dimensions.

**Lemma 7.** *Let $x_1, \ldots, x_{\ell_1}, y_1, \ldots, y_{\ell_2} \in \mathbb{R}^d$ be values that are fully observed. Suppose $a_1, \ldots, a_{\ell_1}, b_1, \ldots, b_{\ell_2}, \beta$ are positive constants. For $z = (z(1), \ldots, z(d)) \in \mathbb{R}^d$ with missing components, define*

$$T(\{z(j) \,:\, j \in U_z\}) = \sum_{i=1}^{\ell_1} a_i \exp\left(-\beta \sum_{j \in U_z} |x_i(j) - z(j)|\right) - \sum_{i=1}^{\ell_2} b_i \exp\left(-\beta \sum_{j \in U_z} |y_i(j) - z(j)|\right)$$

*as a function of the unobserved components of $z$ and let*

$$\mathcal{X} = \{T(\{z(j) \,:\, j \in U_z\}) \,:\, z(i) \in \{x_1(i), \ldots, x_{\ell_1}(i), y_1(i), \ldots, y_{\ell_2}(i)\}, i \in U_z\}.$$

*Then, for any possible imputation $z^*$ of $z$,*

$$T(\{z^*(j) \,:\, j \in U_z\}) \geq \min\{0, \min \mathcal{X}\}, \tag{14}$$

*and*

$$T(\{z^*(j) \,:\, j \in U_z\}) \leq \max\{0, \max \mathcal{X}\}. \tag{15}$$

*Proof.* We will only prove inequality 14 and inequality 15 can be proved following the same method.

Let us prove inequality 14 using mathematical induction. When $|U_z| = 1$, this lemma become Lemma 6, which has already been proved. Suppose when $|U_z| = l - 1$, where $l$ is any integer such that $1 \leq l$, inequality 14 holds. We are going to show when $|U_z| = l$, inequality 14 still holds.

Without loss of generality, let us assume (after relabelling) that $U_z = 1, \ldots, l$. Subsequently, for any given $(z_1^*, \ldots, z_l^*) \in \mathbb{R}^l$, it follows

$$
\begin{aligned}
T(z_1^*, \ldots, z_l^*) &= \sum_{i=1}^{\ell_1} a_i \exp\left(-\beta \sum_{j=1}^{l} |x_i(j) - z_j^*|\right) - \sum_{i=1}^{\ell_2} b_i \exp\left(-\beta \sum_{j=1}^{l} |y_i(j) - z_j^*|\right) \\
&= \sum_{i=1}^{\ell_1} a_i \exp\left(-\beta |x_i(1) - z_1^*|\right) \exp\left(-\beta \sum_{j=2}^{l} |x_i(j) - z_j^*|\right) \\
&\quad - \sum_{i=1}^{\ell_2} b_i \exp\left(-\beta |y_i(1) - z_1^*|\right) \exp\left(-\beta \sum_{j=2}^{l} |y_i(j) - z_j^*|\right).
\end{aligned}
$$

Denote

$$
\begin{aligned}
a_i' &= a_i \exp\left(-\beta |x_i(1) - z_1^*|\right), \ i = 1, \ldots, \ell_1, \\
b_i' &= b_i \exp\left(-\beta |y_i(1) - z_1^*|\right), \ i = 1, \ldots, \ell_2.
\end{aligned}
$$

Consider a new function

$$T'(z_2, \ldots, z_l) = \sum_{i=1}^{\ell_1} a_i' \exp\left(-\beta \sum_{j=2}^{l} |x_i(j) - z_j|\right) - \sum_{i=1}^{\ell_2} b_i' \exp\left(-\beta \sum_{j=2}^{l} |y_i(j) - z_j|\right).$$

Let

$$\mathcal{X}' = \{T'(z_2, \ldots, z_l) | z_j \in \{x_1(j), \ldots, x_{\ell_1}(j), y_1(j), \ldots, y_{\ell_2}(j)\}, j = 2, \ldots, l\}.$$

Then, using the assumption that when $|U_z| = l - 1$, inequality 14 holds, it follows that

$$T'(z_2^*, \ldots, z_l^*) \geq \min\{0, \min \mathcal{X}'\},$$

Notice that

$$
\begin{aligned}
T'(z_2, \ldots, z_l) &= \sum_{i=1}^{\ell_1} a_i' \exp\left(-\beta \sum_{j=2}^{l} |x_i(j) - z_j|\right) - \sum_{i=1}^{\ell_2} b_i' \exp\left(-\beta \sum_{j=2}^{l} |y_i(j) - z_j|\right) \\
&= \sum_{i=1}^{\ell_1} a_i \exp\left(-\beta |x_i(1) - z_1^*|\right) \exp\left(-\beta \sum_{j=2}^{l} |x_i(j) - z_j|\right) \\
&\quad - \sum_{i=1}^{\ell_2} b_i \exp\left(-\beta |y_i(1) - z_1^*|\right) \exp\left(-\beta \sum_{j=2}^{l} |y_i(j) - z_j|\right) \\
&= T(z_1^*, z_2, \ldots, z_l).
\end{aligned}
\tag{16}
$$

Hence,

$$
T'(z_2^*, \ldots, z_l^*) = T(z_1^*, \ldots, z_l^*).
$$

We therefore have

$$
T(z_1^*, \ldots, z_l^*) \geq \min\{0, \min \chi'\}.
\tag{17}
$$

Further, let us denote

$$
(z_2', \ldots, z_l') = \underset{z_i \in \{x_1(i), \ldots, x_{\ell_1}(i), y_1(i), \ldots, y_{\ell_2}(i)\}, i=2,\ldots,l}{\arg\min} T'(z_2, \ldots, z_l).
\tag{18}
$$

That is,

$$
T'(z_2', \ldots, z_l') = \min \mathcal{X}'.
$$

Applying equation 16, it follows

$$
T(z_1^*, z_2', \ldots, z_l') = T'(z_2', \ldots, z_l') = \min \mathcal{X}'.
\tag{19}
$$

Denote

$$
a_i'' = a_i \exp\left(-\beta \sum_{j=2}^{l} |x_i(j) - z_j'|\right), \quad i = 1, \ldots, \ell_1,
$$

$$
b_i'' = b_i \exp\left(-\beta \sum_{j=2}^{l} |y_i(j) - z_j'|\right) \quad i = 1, \ldots, \ell_2.
$$

Consider a new function

$$
T''(z_1) = \sum_{i=1}^{\ell_1} a_i'' \exp\left(-\beta |x_i(1) - z_1|\right) - \sum_{i=1}^{\ell_1} b_i'' \exp\left(-\beta |y_i(1) - z_1|\right).
$$

Let

$$
\mathcal{X}'' = \{T''(z_1) | z_1 \in \{x_1(1), \ldots, x_{\ell_1}(1), y_1(1), \ldots, y_{\ell_2}(1)\}\}.
$$

Then, using the result that when $|U_z| = 1$, equation 14 holds, it follows

$$
T''(z_1^*) \geq \min\{0, \min \mathcal{X}''\}.
\tag{20}
$$

Notice that

$$
\begin{aligned}
T''(z_1) &= \sum_{i=1}^{\ell_1} a_i'' \exp\left(-\beta |x_i(1) - z_1|\right) - \sum_{i=1}^{\ell_1} b_i'' \exp\left(-\beta |y_i(1) - z_1|\right) \\
&= \sum_{i=1}^{\ell_1} a_i \exp\left(-\beta \sum_{j=2}^{l} |x_i(j) - z_j'|\right) \exp\left(-\beta |x_i(1) - z_1|\right) \\
&\quad - \sum_{i=1}^{\ell_1} b_i \exp\left(-\beta \sum_{j=2}^{l} |y_i(j) - z_j'|\right) \exp\left(-\beta |y_i(1) - z_1|\right) \\
&= T(z_1, z_2', \dots, z_l').
\end{aligned}
$$

According to equation 19,

$$
T''(z_1^*) = \min \mathcal{X}'.
$$

Notice that since equation 20 holds, we have

$$
\min \mathcal{X}' \geq \min\{0, \min \mathcal{X}''\},
$$

put which back into equation 17, we further have

$$
T(z_1^*, \dots, z_l^*) \geq \min\{0, \min \mathcal{X}'\} \geq \min\{0, \min \mathcal{X}''\}.
$$

Notice that

$$
\begin{aligned}
\min \mathcal{X}'' &= \min\{T''(z_1) | z_1 \in \{x_1(1), \dots, x_{\ell_1}(1), y_1(1), \dots, y_{\ell_2}(1)\}\} \\
&= \min\{T(z_1, z_2', \dots, z_l') | z_1 \in \{x_1(1), \dots, x_{\ell_1}(1), y_1(1), \dots, y_{\ell_2}(1)\}\},
\end{aligned}
$$

where

$$
z_i' \in \{x_1(i), \dots, x_{\ell_1}(i), y_1(i), \dots, y_{\ell_2}(i)\}, i = 2, \dots, l
$$

according to its definition in equation 18. Hence,

$$
\min \mathcal{X}'' \geq \min \chi \implies T(z_1^*, \dots, z_l^*) \geq \min\{0, \min \mathcal{X}\},
$$

which proves inequality 14. □

### A.4 Proof of Lemma 4

**Lemma 8.** *Following the notation and definitions in Lemma 3, denote*

$$
\tilde{x}_i(j) = \max\{|x_i(j) - x_1(j)|, \dots, |x_i(j) - x_{\ell_1}(j)|, |x_i(j) - y_1(j)|, \dots, |x_i(j) - y_{\ell_2}(j)|\}
$$

*for any $i \in \{1, \dots, \ell_1\}, j \in U_z$, and denote*

$$
\tilde{y}_i(j) = \max\{|y_i(j) - x_1(j)|, \dots, |y_i(j) - x_{\ell_1}(j)|, |y_i(j) - y_1(j)|, \dots, |y_i(j) - y_{\ell_2}(j)|\}
$$

*for any $i \in \{1, \dots, \ell_2\}, j \in U_z$. Subsequently,*

$$
\min \mathcal{X} \geq \sum_{i=1}^{\ell_1} a_i \exp\left(-\beta \sum_{j \in U_z} \tilde{x}_i(j)\right) - \sum_{i=1}^{\ell_2} b_i, \tag{21}
$$

$$
\max \mathcal{X} \leq \sum_{i=1}^{\ell_1} a_i - \sum_{i=1}^{\ell_2} b_i \exp\left(-\beta \sum_{j \in U_z} \tilde{y}_i(j)\right). \tag{22}
$$

*Proof.* We will only prove inequality 21 and inequality 22 can be proved similarly.

Without loss of generality, let us assume (after relabelling) $U_z = 1, \ldots, l$, where $l = |U_z|$. Subsequently, let

$$(z_1^*, \ldots, z_l^*) = \underset{\{z_i \in \{x_1(i), \ldots, x_{\ell_1}(i), y_1(i), \ldots, y_{\ell_2}(i)\}, i=1, \ldots, l\}}{\arg \min} T(z_1, \ldots, z_l).$$

That is,

$$T(z_1^*, \ldots, z_l^*) = \min \mathcal{X}.$$

Notice that for any $j \in \{1, \ldots, l\}$,

$$z_j^* \in \{x_1(j), \ldots, x_{\ell_1}(j), y_1(j), \ldots, y_{\ell_2}(j)\}$$

according to its definition. Hence, for any $i \in \{1, \ldots, \ell_1\}$,

$$|z_j^* - x_i(j)| \le \max\{|x_i(j) - x_1(j)|, \ldots, |x_i(j) - x_{\ell_1}(j)| = \tilde{x}_i(j),$$

following which

$$\sum_{j=1}^{l} \tilde{x}_i(j) \ge \sum_{j=1}^{l} |x_i(j) - z_j^*|, \ i = \{1, \ldots, \ell_1\}$$

$$\implies -\beta \sum_{j=1}^{l} \tilde{x}_i(j) \le -\beta \sum_{j=1}^{l} |x_i(j) - z_j^*|, \ i = \{1, \ldots, \ell_1\}.$$

Subsequently,

$$T(z_1^*, \ldots, z_l^*) = \sum_{i=1}^{\ell_1} a_i \exp\left(-\beta \sum_{j=1}^{l} |x_i(j) - z_j^*|\right) - \sum_{i=1}^{\ell_2} b_i \exp\left(-\beta \sum_{j=1}^{l} |y_i(j) - z_j^*|\right)$$

$$\ge \sum_{i=1}^{\ell_1} a_i \exp\left(-\beta \sum_{j=1}^{l} \tilde{x}_i(j)\right) - \sum_{i=1}^{\ell_2} b_i \exp\left(-\beta \sum_{j=1}^{l} |y_i(j) - z_j^*|\right)$$

$$\ge \sum_{i=1}^{\ell_1} a_i \exp\left(-\beta \sum_{j=1}^{l} \tilde{x}_i(j)\right) - \sum_{i=1}^{\ell_2} b_i,$$

which proves inequality 21. $\qquad \square$

### A.5 Proof of Theorem 1

We now prove Theorem 1, which bounds the $\mathrm{MMD}_u^2(X, Y)$ test statistic when data are univariate.

**Theorem 5.** *Suppose $X = \{x_1, \ldots, x_{n_1}\}$ and $Y = \{y_1, \ldots, y_{n_2}\}$ are univariate real values. Assume $x_1, \ldots, x_{m_1}, y_1, \ldots, y_{m_2}$ are unobserved. Let $k$ denote the Laplacian kernel and define*

$$T_1(z) = c_1 \sum_{j=m_1+1}^{n_1} k(z, x_j) - c_3 \sum_{j=m_2+1}^{n_2} k(z, y_j),$$

$$T_2(z) = c_2 \sum_{j=m_2+1}^{n_2} k(z, x_j) - c_3 \sum_{j=m_1+1}^{n_1} k(z, y_j),$$

*where $c_1 = \frac{2}{n_1(n_1-1)}, c_2 = \frac{2}{n_2(n_2-1)}$ and $c_3 = \frac{2}{n_1 n_2}$. Further, let*

$$S_1 := \{0, T_1(x_{m_1+1}), \ldots, T_1(x_{n_1}), T_1(y_{m_2+1}), \ldots, T_1(y_{n_2})\},$$
$$S_2 := \{0, T_2(x_{m_2+1}), \ldots, T_2(x_{n_2}), T_2(y_{m_1+1}), \ldots, T_2(y_{n_1})\}.$$

*Then, the $MMD_u^2(X,Y)$ using Laplacian kernel $k$ is bounded, over all possible values of $x_1, \ldots, x_{m_1}$, $y_1, \ldots, y_{m_2}$, as follows:*

$$\frac{m_1(m_1-1)}{n_1(n_1-1)} + \frac{m_2(m_2-1)}{n_2(n_2-1)} + m_1 \max S_1 + m_2 \max S_2 + A_2 > MMD_u^2(X,Y),$$

$$MMD_u^2(X,Y) > A_2 + m_1 \min S_1 + m_2 \min S_2 - \frac{2}{n_1 n_2} m_1 m_2,$$

*where $A_2$ is defined in lemma 1.*

*Proof.* To start, according to Lemma 1,

$$\mathrm{MMD}_u^2(X,Y) = A_1 + A_2 + A_3 + A_4,$$

where

$$A_1 = c_1 \sum_{i=1}^{m_1} \sum_{j=i+1}^{m_1} k(x_i,x_j) + c_2 \sum_{i=1}^{m_2} \sum_{j=i+1}^{m_2} k(y_i,y_j) - c_3 \sum_{i=1}^{m_1} \sum_{j=1}^{m_2} k(x_i,y_j),$$

$$A_2 = c_1 \sum_{i=m_1+1}^{n_1-1} \sum_{j=i+1}^{n_1} k(x_i,x_j) + c_2 \sum_{i=m_2+1}^{n_2-1} \sum_{j=i+1}^{n_2} k(y_i,y_j) - c_3 \sum_{i=m_1+1}^{n_1} \sum_{j=m_2+1}^{n_2} k(x_i,y_j),$$

$$A_3 = c_1 \sum_{i=1}^{m_1} \sum_{j=m_1+1}^{n_1} k(x_i,x_j) - c_3 \sum_{i=1}^{m_1} \sum_{j=m_2+1}^{n_2} k(x_i,y_j),$$

$$A_4 = c_2 \sum_{i=1}^{m_2} \sum_{j=m_2+1}^{n_2} k(y_i,y_j) - c_3 \sum_{i=m_1+1}^{n_1} \sum_{j=1}^{m_2} k(x_i,y_j),$$

and $c_1 = \frac{2}{n_1(n_1-1)}, c_2 = \frac{2}{n_2(n_2-1)}$ and $c_3 = \frac{2}{n_1 n_2}$.

Following Equation equation 2, it is provided

$$\frac{m_1(m_1-1)}{n_1(n_1-1)} + \frac{m_2(m_2-1)}{n_2(n_2-1)} > A_1 > -\frac{2}{n_1 n_2} m_1 m_2.$$

Further, using Lemma 2, it follows

$$m_1 \max S_1 \geq A_3 \geq m_1 \min S_1,$$

and

$$m_2 \max S_2 \geq A_4 \geq m_2 \min S_2,$$

which concludes our proof. □

In order to compute the bounds of $\mathrm{MMD}_u^2(X,Y)$ for univariate samples with missing data using Theorem 5, we need to compute $\min S_1, \max S_1, \min S_2$ and $\max S_2$. Since $|S_1| = n_1 - m_1$ and $|S_2| = n_2 - m_2$. The computation complexity is of order $O(n_1 + n_2)$.

### A.6 Proof of Theorem 2

This subsection proves Theorem 2, which bounds the $\mathrm{MMD}_u^2(X,Y)$ test statistic for multivariate data. To start, we prove the following result regarding the incomplete Laplacian kernel $\mathcal{K}$ defined in Definition 3

**Lemma 5.** *Suppose $x$ and $y$ are two samples of $\mathbb{R}^d$ real values and assume not all dimensions of values of $x$ and $y$ are observed. Denote $[d] = \{1, \ldots, d\}$. Let $U_x$ be a set includes all unobserved dimensions in $x$ and $U_y$ be a set includes all unobserved dimensions in $y$. Let $\mathcal{K}$ denote the incomplete Laplacian kernel defined in Definition 3. Subsequently, it follows*

$$\mathcal{K}(x,y) \geq k(x,y) \geq 0.$$

*Proof.* According to the definition of the Laplacian kernel, we have

$$k(x, y) = \exp\left(-\beta \sum_{l=1}^{d} |x(l) - y(l)|\right) \geq 0.$$

Also, notice that

$$\sum_{l \in [d] \setminus (U_x \cup U_y)} |x(l) - y(l)| \leq \sum_{l=1}^{d} |x(l) - y(l)|.$$

Then, since $\beta > 0$,

$$-\beta \sum_{i \in [d] \setminus (U_x \cup U_y)} |x(i) - y(i)| \geq -\beta \sum_{l=1}^{d} |x(l) - y(l)|.$$

According to the definition of incomplete Laplacian kernel, we therefore have

$$\mathcal{K}(x, y) = \exp\left(-\beta \sum_{l \in [d] \setminus (U_x \cup U_y)} |x(l) - y(l)|\right)$$

$$\geq \exp\left(-\beta \sum_{l=1}^{d} |x(l) - y(l)|\right)$$

$$= k(x, y),$$

which completes our proof. $\qquad\square$

Recall that in Lemma 1, it is shown that $\mathrm{MMD}_u^2(X, Y)$ can be decomposed into four parts $A_1, A_2, A_3$ and $A_4$ with part $A_1$ including incomplete samples only, $A_2$ including complete samples only, and $A_3, A_4$ mixed with both complete and incomplete samples.

Lemma 5 allows us to bound term $A_1$ by bounding any two incompletely observed samples separately. Let $U_{x_i}$ be a set including all unobserved dimensions in $x_i$, where $1 \leq i \leq m_1$. Similarly, let $U_{y_i}$ be a set including all unobserved dimensions in $y_i$, where $1 \leq i \leq m_2$. Subsequently, according to Lemma 5, it can be seen that $\sum_{i=1}^{m_1} \sum_{j=i+1}^{m_1} \mathcal{K}(x_i, x_j) \geq \sum_{i=1}^{m_1} \sum_{j=i+1}^{m_1} k(x_i, x_j) > 0$, $\sum_{i=1}^{m_2} \sum_{j=i+1}^{m_2} \mathcal{K}(y_i, y_j) \geq \sum_{i=1}^{m_2} \sum_{j=i+1}^{m_2} k(y_i, y_j) > 0$, and $\sum_{i=1}^{m_1} \sum_{j=1}^{m_2} \mathcal{K}(x_i, y_j) \geq \sum_{i=1}^{m_1} \sum_{j=1}^{m_2} k(x_i, y_j) > 0$. Then, according to the definition of $A_1$ in Lemma 1, it follows

$$c_1 \sum_{i=1}^{m_1} \sum_{j=i+1}^{m_1} \mathcal{K}(x_i, x_j) + c_2 \sum_{i=1}^{m_2} \sum_{j=i+1}^{m_2} \mathcal{K}(y_i, y_j) > A_1 > -c_3 \sum_{i=1}^{m_1} \sum_{j=1}^{m_2} \mathcal{K}(x_i, y_j), \tag{23}$$

where $c_1 = \frac{2}{n_1(n_1-1)}, c_2 = \frac{2}{n_2(n_2-1)}$ and $c_3 = \frac{2}{n_1 n_2}$. We now bound the terms $A_3$ and $A_4$. Recall that the term $A_3$ can be rewritten as equation 3 in both univariate and multivariate cases. For any $x_i$ where $1 \leq i \leq m_1$, let $U_{x_i}$ be a set including all unobserved dimensions in $x_i$ and let us introduce the function

$$T_1(x_i(l); l \in U_{x_i}) = c_1 \sum_{j=m_1+1}^{n_1} k(x_i, x_j) - c_3 \sum_{j=m_2+1}^{n_2} k(x_i, y_j), \ i \in \{1, \ldots, m_1\} \tag{24}$$

where $c_1 = \frac{2}{n_1(n_1-1)}$ and $c_3 = \frac{2}{n_1 n_2}$. Then, $A_3$ can be computed as

$$A_3 = \sum_{i=1}^{m_1} T_1(x_i(l); l \in U_{x_i}).$$

Subsequently, by combining Lemma 7 and Lemma 8, we have our main result:

**Theorem 6.** *Suppose* $X = \{x_1, \ldots, x_{n_1}\}$ *and* $Y = \{y_1, \ldots, y_{n_2}\}$ *are samples of* $\mathbb{R}^d$ *real values. Assume* $x_1, \ldots, x_{m_1}, y_1, \ldots, y_{m_2}$ *are data that are missing, or not completely observed. Subsequently, the* $MMD_u^2(X, Y)$ *using Laplacian kernel* $k$ *is bounded, over all possible imputations of* $x_1, \ldots, x_{m_1}, y_1, \ldots, y_{m_2}$*, as follows:*

$$c_1 \sum_{i=1}^{m_1} \sum_{j=i+1}^{m_1} \mathcal{K}(x_i, x_j) + c_2 \sum_{i=1}^{m_2} \sum_{j=i+1}^{m_2} \mathcal{K}(y_i, y_j) + A_2 + C_1 + C_3 > MMD_u^2(X, Y),$$

$$MMD_u^2(X, Y) > -c_3 \sum_{i=1}^{m_1} \sum_{j=1}^{m_2} \mathcal{K}(x_i, y_j) + A_2 + C_2 + C_4,$$

*where* $\mathcal{K}$ *is the incomplete Laplacian kernel defined in Definition 3,* $A_2$ *is a constant of observed data defined in Lemma 1, and* $c_1, c_2, c_3$ *and* $C_1, C_2, C_3, C_4$ *are also constants of the observed data defined in Definition 4.*

*Proof.* In Lemma 1, it is proved that

$$\text{MMD}_u^2(X, Y) = A_1 + A_2 + A_3 + A_4,$$

where

$$A_1 = c_1 \sum_{i=1}^{m_1} \sum_{j=i+1}^{m_1} k(x_i, x_j) + c_2 \sum_{i=1}^{m_2} \sum_{j=i+1}^{m_2} k(y_i, y_j) - c_3 \sum_{i=1}^{m_1} \sum_{j=1}^{m_2} k(x_i, y_j),$$

$$A_2 = c_1 \sum_{i=m_1+1}^{n_1-1} \sum_{j=i+1}^{n_1} k(x_i, x_j) + c_2 \sum_{i=m_2+1}^{n_2-1} \sum_{j=i+1}^{n_2} k(y_i, y_j) - c_3 \sum_{i=m_1+1}^{n_1} \sum_{j=m_2+1}^{n_2} k(x_i, y_j),$$

$$A_3 = c_1 \sum_{i=1}^{m_1} \sum_{j=m_1+1}^{n_1} k(x_i, x_j) - c_3 \sum_{i=1}^{m_1} \sum_{j=m_2+1}^{n_2} k(x_i, y_j),$$

$$A_4 = c_2 \sum_{i=1}^{m_2} \sum_{j=m_2+1}^{n_2} k(y_i, y_j) - c_3 \sum_{i=m_1+1}^{n_1} \sum_{j=1}^{m_2} k(x_i, y_j),$$

and $c_1 = \frac{2}{n_1(n_1-1)}, c_2 = \frac{2}{n_2(n_2-1)}$ and $c_3 = \frac{2}{n_1 n_2}$.

According to inequality 23,

$$c_1 \sum_{i=1}^{m_1} \sum_{j=i+1}^{m_1} \mathcal{K}(x_i, x_j) + c_2 \sum_{i=1}^{m_2} \sum_{j=i+1}^{m_2} \mathcal{K}(y_i, y_j) > A_1 > -c_3 \sum_{i=1}^{m_1} \sum_{j=1}^{m_2} \mathcal{K}(x_i, y_j). \tag{25}$$

Notice that

$$A_3 = \sum_{i=1}^{m_1} \left( c_1 \sum_{j=m_1+1}^{n_1} k(x_i, x_j) - c_3 \sum_{j=m_2+1}^{n_2} k(x_i, y_j) \right).$$

Let us introduce the function

$$T_1(x_i(l); l \in U_{x_i}) = c_1 \sum_{j=m_1+1}^{n_1} k(x_i, x_j) - c_3 \sum_{j=m_2+1}^{n_2} k(x_i, y_j), \ i \in \{1, \ldots, m_1\}.$$

Subsequently, $A_3$ can be computed as

$$A_3 = \sum_{i=1}^{m_1} T_1(x_i(l); l \in U_{x_i}). \tag{26}$$

Denote

$$\mathcal{X}_{x_i} = \{T_1(x_i(l); l \in U_{x_i}) | x_l \in \{x_{m_1+1}(l), \ldots, x_{n_1}(l), y_{m_2+1}(l), \ldots, y_{n_2}(l)\}, l \in U_{x_i}\},$$

for any $i = 1, \ldots, m_1$. Let $[d] = \{1, \ldots, d\}$. Notice that

$$T_1(x_i(l); l \in U_{x_i}) = \sum_{j=m_1+1}^{n_1} a'_{i,j} \exp\left(-\beta \sum_{l \in U_{x_i}} |x_i(l) - x_j(l)|\right) -$$

$$\sum_{j=m_2+1}^{n_2} b'_{i,j} \exp\left(-\beta \sum_{l \in U_{x_i}} |x_i(l) - y_j(l)|\right),$$

where

$$a'_{i,j} = c_1 \exp\left(-\beta \sum_{l \in [d]\backslash U_{x_i}} |x_i(l) - x_j(l)|\right),$$

$$b'_{i,j} = c_3 \exp\left(-\beta \sum_{l \in [d]\backslash U_{x_i}} |x_i(l) - y_j(l)|\right).$$

Then, according to Lemma 3, it follows

$$\max\{0, \max \mathcal{X}_{x_i}\} \geq T_1(x_i(l); l \in U_{x_i}) \geq \min\{0, \min \mathcal{X}_{x_i}\}, i = 1, \ldots, m_1. \tag{27}$$

According to Lemma 4,

$$\min \mathcal{X}_{x_i} \geq \sum_{j=m_1+1}^{n_1} a'_{i,j} \exp\left(-\beta \sum_{l \in U_{x_i}} \tilde{x}_i(l)\right) - \sum_{j=m_2+1}^{n_2} b'_{i,j},$$

$$\max \mathcal{X}_{x_i} \leq \sum_{j=m_1+1}^{n_1} a'_{i,j} - \sum_{j=m_2+1}^{n_2} b'_{i,j} \exp\left(-\beta \sum_{l \in U_{x_i}} \tilde{y}_j(l)\right).$$

Notice that according to the definition of incomplete kernel in Definition 3,

$$a'_{i,j} = c_1 \exp\left(-\beta \sum_{l \in [d]\backslash U_{x_i}} |x_i(l) - x_j(l)|\right) = c_1 \mathcal{K}(x_i, x_j),$$

$$b'_{i,j} = c_3 \exp\left(-\beta \sum_{l \in [d]\backslash U_{x_i}} |x_i(l) - y_j(l)|\right) = c_3 \mathcal{K}(x_i, y_j).$$

Hence, for any $i = 1, \ldots, m_1$,

$$\min \mathcal{X}_{x_i} \geq \sum_{j=m_1+1}^{n_1} c_1 \mathcal{K}(x_i, x_j) \exp\left(-\beta \sum_{l \in U_{x_i}} \tilde{x}_i(l)\right) - \sum_{j=m_2+1}^{n_2} c_3 \mathcal{K}(x_i, y_j),$$

$$\max \mathcal{X}_{x_i} \leq \sum_{j=m_1+1}^{n_1} c_1 \mathcal{K}(x_i, x_j) - \sum_{j=m_2+1}^{n_2} c_3 \mathcal{K}(x_i, y_j) \exp\left(-\beta \sum_{l \in U_{x_i}} \tilde{y}_j(l)\right).$$

According to equation 27, we then have

$$T_1(x_i(l); l \in U_{x_i}) \geq \min\{0, \min \mathcal{X}_{x_i}\}$$

$$\geq \min\left\{0, \sum_{j=m_1+1}^{n_1} c_1 \mathcal{K}(x_i, x_j) \exp\left(-\beta \sum_{l \in U_{x_i}} \tilde{x}_i(l)\right) - \sum_{j=m_2+1}^{n_2} c_3 \mathcal{K}(x_i, y_j)\right\},$$

and

$$T_1(x_i(l); l \in U_{x_i}) \leq \max\{0, \max \mathcal{X}_{x_i}\}$$

$$\leq \max \left\{ 0, \sum_{j=m_1+1}^{n_1} c_1 \mathcal{K}(x_i, x_j) - \sum_{j=m_2+1}^{n_2} c_3 \mathcal{K}(x_i, y_j) \exp\left( -\beta \sum_{l \in U_{x_i}} \tilde{y}_j(l) \right) \right\}.$$

Following equation 26, then

$$C_1 \geq A_3 \geq C_2.$$

Using the similar method, it can be prove that

$$C_3 \geq A_4 \geq C_4.$$

Combining with equation 27, we conclude our result. □

For computing bounds of $\text{MMD}_u^2(X, Y)$ using Theorem 6, we need to compute $\tilde{x}_i(l)$ for any $i \in \{m_1 + 1, \ldots, n_1\}, l \in \{1, \ldots, d\}$, and compute $\tilde{y}_i(l)$ for any $i \in \{m_2 + 1, \ldots, n_2\}, l \in \{1, \ldots, d\}$. Notice that for any $\tilde{x}_i(l)$, we have

$$|\{|x_i(l) - x_{m_1+1}(l)|, \ldots, |x_i(l) - x_{n_1}(l)|, |x_i(l) - y_{m_2+1}(l)|, \ldots, |x_i(l) - y_{n_2}(l)|\}|$$
$$= n_1 - m_1 + n_2 - m_2.$$

Hence, computing all $\tilde{x}_i(l)$ is of computational order at most $O(dn_1(n_1 + n_2))$. Similarly, computing all $\tilde{y}_i(l)$ is of computational complexity order at most $O(dn_2(n_1 + n_2))$. After computing $\tilde{x}_i(l)$ for any $i \in \{m_1 + 1, \ldots, n_1\}, l \in \{1, \ldots, d\}$, and $\tilde{y}_i(l)$ for any $i \in \{m_2 + 1, \ldots, n_2\}, l \in \{1, \ldots, d\}$, $C_1, C_2, C_3$ and $C_4$ can be computed with computational complexity order at most $O((n_1 + n_2))$. Overall, the computational complexity order of computing bounds of $\text{MMD}_u^2(X, Y)$ using Theorem 6, is $O((n_1 + n_2))$.

### A.7 Proof of Theorem 3

This section provides bounds of $p$-value of MMD test statistic based on the bounds of MMD derived in Section 3, and the permutation test.

**Theorem 2.** *Suppose $X = \{x_1, \ldots, x_{n_1}\}$ and $Y = \{y_1, \ldots, y_{n_2}\}$ are samples of $\mathbb{R}^d$ real value. Let $(\sigma^{(1)}, \ldots, \sigma^{(B)})$ be $B$ i.i.d. random samplings, each being random permutation of $\{1, \ldots, n_1 + n_2\}$ and denoted as $\sigma^{(i)} = (\sigma^{(i)}(1), \ldots, \sigma^{(i)}(n_1 + n_2))$, $i = 1, \ldots, B$. Subsequently, let $z_1 = x_1, \ldots, z_{n_1} = x_{n_1}, z_{n_1+1} = y_1, \ldots, z_{n_1+n_2} = y_{n_2}$ and for any $i = 1, \ldots, B$, denote*

$$X_{\sigma^{(i)}} = \{z_{\sigma^{(i)}(1)}, \ldots, z_{\sigma^{(i)}(n_1)}\}, Y_{\sigma^{(i)}} = \{z_{\sigma^{(i)}(n_1+1)}, \ldots, z_{\sigma^{(i)}(n_1+n_2)}\},$$

*and define $p$ according to equation 5. Suppose further*

$$\underline{MMD}_u^2(X, Y) \leq MMD_u^2(X, Y), \ MMD_u^2(X_{\sigma^{(i)}}, Y_{\sigma^{(i)}}) \leq \overline{MMD}_u^2(X_{\sigma^{(i)}}, Y_{\sigma^{(i)}}), \ i = 1, \ldots, B.$$

*Define*

$$\overline{p} = \frac{1}{B+1}\left( 1 + \sum_{i=1}^{B} \mathbb{I}(\overline{MMD}_u^2(X_{\sigma^{(i)}}, Y_{\sigma^{(i)}}) \geq \underline{MMD}_u^2(X, Y)) \right).$$

*Then, we must have $\overline{p} \geq p$.*

*Proof.* Notice that for any $i \in \{1, \ldots, B\}$, we have

$$\mathbb{I}(\text{MMD}_u^2(X_{\sigma^{(i)}}, Y_{\sigma^{(i)}}) \geq \text{MMD}_u^2(X, Y)) = 1$$
$$\implies \text{MMD}_u^2(X_{\sigma^{(i)}}, Y_{\sigma^{(i)}}) \geq \text{MMD}_u^2(X, Y).$$

Since

$$\text{MMD}_u^2(X,Y) \geq \underline{\text{MMD}}_u^2(X,Y), \ \overline{\text{MMD}}_u^2(X_{\sigma(i)}, Y_{\sigma(i)}) \geq \text{MMD}_u^2(X_{\sigma(i)}, Y_{\sigma(i)}), \ i = 1, \ldots, B,$$

it then follows

$$\overline{\text{MMD}}_u^2(X_{\sigma(i)}, Y_{\sigma(i)}) \geq \text{MMD}_u^2(X_{\sigma(i)}, Y_{\sigma(i)}) \geq \underline{\text{MMD}}_u^2(X,Y),$$
$$\implies \mathbb{I}(\overline{\text{MMD}}_u^2(X_{\sigma(i)}, Y_{\sigma(i)}) \geq \underline{\text{MMD}}_u^2(X,Y)) = 1.$$

for any $i \in \{1, \ldots, B\}$.

If, for any $i \in \{1, \ldots, B\}$,

$$\mathbb{I}(\text{MMD}_u^2(X_{\sigma(i)}, Y_{\sigma(i)}) \geq \text{MMD}_u^2(X,Y)) = 0,$$

then we have

$$\mathbb{I}(\overline{\text{MMD}}_u^2(X_{\sigma(i)}, Y_{\sigma(i)}) \geq \underline{\text{MMD}}_u^2(X,Y)) = 0 \text{ or } 1.$$

Hence, for any $i \in \{1, \ldots, B\}$, we have

$$\mathbb{I}(\overline{\text{MMD}}_u^2(X_{\sigma(i)}, Y_{\sigma(i)}) \geq \underline{\text{MMD}}_u^2(X,Y)) \geq \mathbb{I}(\text{MMD}_u^2(X_{\sigma(i)}, Y_{\sigma(i)}) \geq \text{MMD}_u^2(X,Y)).$$

According to the definitions of $p$ and $\overline{p}$, we then must have $\overline{p} \geq p$.

$\square$

## A.8  Proof of Theorem 4

This section provides bounds of $p$-value of MMD test statistic, based on the bounds of MMD derived in Section 3, and the normality approximation proposed in Gao & Shao (2023).

**Theorem 3.** *Suppose* $X = \{X_1, \ldots, X_{n_1}\}, Y = \{Y_1, \ldots, Y_{n_2}\} \in \mathbb{R}^d$, *and assume* $X_1, \ldots, X_{m_1}$ *and* $Y_1, \ldots, Y_{m_2}$ *are not observed. Let* $\mathcal{K}$ *denote the incomplete Laplacian kernel defined in Definition 3. Then, we have* $\overline{\mathcal{V}}_{n,m}^{\mathcal{K}*} \geq \mathcal{V}_{n,m}^{k*}$, *Further, suppose we have* $\underline{MMD}_u^2(X,Y)$ *such that*

$$0 \leq \underline{MMD}_u^2(X,Y) \leq MMD_u^2(X,Y),$$

*we have*

$$\frac{\underline{MMD}_u^2(X,Y)}{\sqrt{c_{n,m} \overline{\mathcal{V}}_{n,m}^{\mathcal{K}*}}} \leq \frac{MMD_u^2(X,Y)}{\sqrt{c_{n,m} \mathcal{V}_{n,m}^{k*}}},$$

*where* $c_{n,m}$ *and* $\mathcal{V}_{n,m}^{k*}$ *are defined in equation 6, and equation 7, separately, and* $\overline{\mathcal{V}}_{n,m}^{\mathcal{K}*}$ *is defined in Definition 5.*

*Proof.* To start, recall that it is proved in Lemma 5 that

$$\mathcal{K}(x,y) \geq k(x,y) > 0,$$

where $x, y$ are potentially incompletely observed $\mathbb{R}^d$ samples. Hence, for any $1 \leq s, t \leq N$, we have

$$\overline{a}_{s,t}^k \geq a_{s,t}^k \geq \underline{a}_{\cdot,t}^k,$$

which then gives

$$\overline{A}_{s,t}^{\mathcal{K}*} \geq A_{s,t}^{k*},$$

where $A_{s,t}^{k*}$ is defined in Proposition 10 in Gao & Shao (2023) as

$$A_{s,t}^{k*} := a_{s,t}^k - a_{\cdot,t}^k - a_{s,\cdot}^k + a_{\cdot,\cdot}^k.$$

Subsequently,

$$\overline{\mathcal{V}}_{n,m}^{\mathcal{K}*} \geq \mathcal{V}_{n,m}^{k*},$$

where $\mathcal{V}_{n,m}^{k*}$ is defined in Proposition 10 in Gao & Shao (2023) as

$$\mathcal{V}_{n,m}^{k*} = \frac{1}{N(N-3)} \sum_{s \neq t} \left(A_{s,t}^{k*}\right)^2 - \frac{1}{(N-1)(N-3)}.$$

Further, suppose

$$0 \leq \underline{\mathrm{MMD}}_u^2(X,Y) \leq \mathrm{MMD}_u^2(X,Y).$$

Then, we must have

$$\frac{\underline{\mathrm{MMD}}_u^2(X,Y)}{\sqrt{c_{n,m}\overline{\mathcal{V}}_{n,m}^{\mathcal{K}*}}} \leq \frac{\mathrm{MMD}_u^2(X,Y)}{\sqrt{c_{n,m}\mathcal{V}_{n,m}^{k*}}},$$

which completes our proof. $\qquad\square$

# B  Simulation Details

## B.1  Computing Environment and Resources

The experiments were run on an high performance computing cluster with 325 compute nodes, each equipped with 2x AMD EPYC 7742 processors (128 cores, 1TB RAM per node). Each job utilized 32 cores and 64 GB of memory, with compute times varying from a couple of hours to 2-3 days, based on the samples sizes and dimensions of data of the job.

## B.2  Simulation Settings

**Median Heuristic.**  All experiments in Section 5 use median heuristic (Gretton et al., 2012a) for the parameter $\beta$ of the Laplacian kernel. This heuristic first calculates the median of all pairwise differences, i.e.

$$m = \text{med}\{\|z - z'\| \,:\, z, z' \in \{x_1, \ldots, x_{n_1}, y_1, \ldots, y_{n_2}\}, z \neq z'\},$$

and then sets $\beta = 1/m$. For the Laplacian kernel, the norm is the 1-norm. This is a popular method of setting $\beta$, and it generally works well (Gretton et al., 2012a). For case deletion and the proposed methods, only completely observed data will be used for computing $\beta$. For mean and hot deck imputation, the $\beta$ will be computed using imputed data.

**Permutations.**  Permutation tests Good (2005); Schrab et al. (2023) are used for case deletion, mean imputation, and hot deck imputation, with the number of permutations set to $B = 100$. MMD-Perm also uses $B = 100$ permutations.

## B.3  Other missing data methods

**MissForest.**  MissForest Stekhoven & Bühlmann (2012) is a nonparametric imputation method based on random forests. It iteratively imputes missing values by training a random forest on observed data for each variable with missing entries, using the other variables as predictors. The process repeats until the imputed values converge or a stopping criterion is met. In our experiments, we evaluate this method using the default parameters recommended in Stekhoven & Stekhoven (2013).

**Wilcoxon-Mann-Whitney test with missing data method (WMWM).**  WMWM Zeng et al. (2024)is a two sample testing with missing data method based on the Wilcoxon-Mann-Whitney test statistic. This method rejects the null hypothesis when all possible imputed test statistics are significant. Since WMWM is based on the Wilcoxon-Mann-Whitney test statistic, it is applicable only for univariate data and can is restricted to detect mean value shift.

**Case deletion.**  For univariate data, case deletion uses only the observed data for testing. For multivariate data, only data that are completely observed are used.

**Mean imputation.**  For univariate data, mean imputation replaces missing values with the mean of the observed samples in the same group, $X$ or $Y$. For multivariate data, the missing components of an incomplete vector are imputed with the mean of the observed components of that same vector.

**Hot deck imputation.**  For univariate data, hot deck imputation imputes missing values with randomly selected observed values (with replacement) from the same group, $X$ or $Y$. For multivariate data, the missing components of an incomplete vector are imputed with randomly selected observed components (with replacement) from that same vector.

## B.4 Examples of MNIST dataset.

In this section, we present examples of MNIST images used in Section 5.

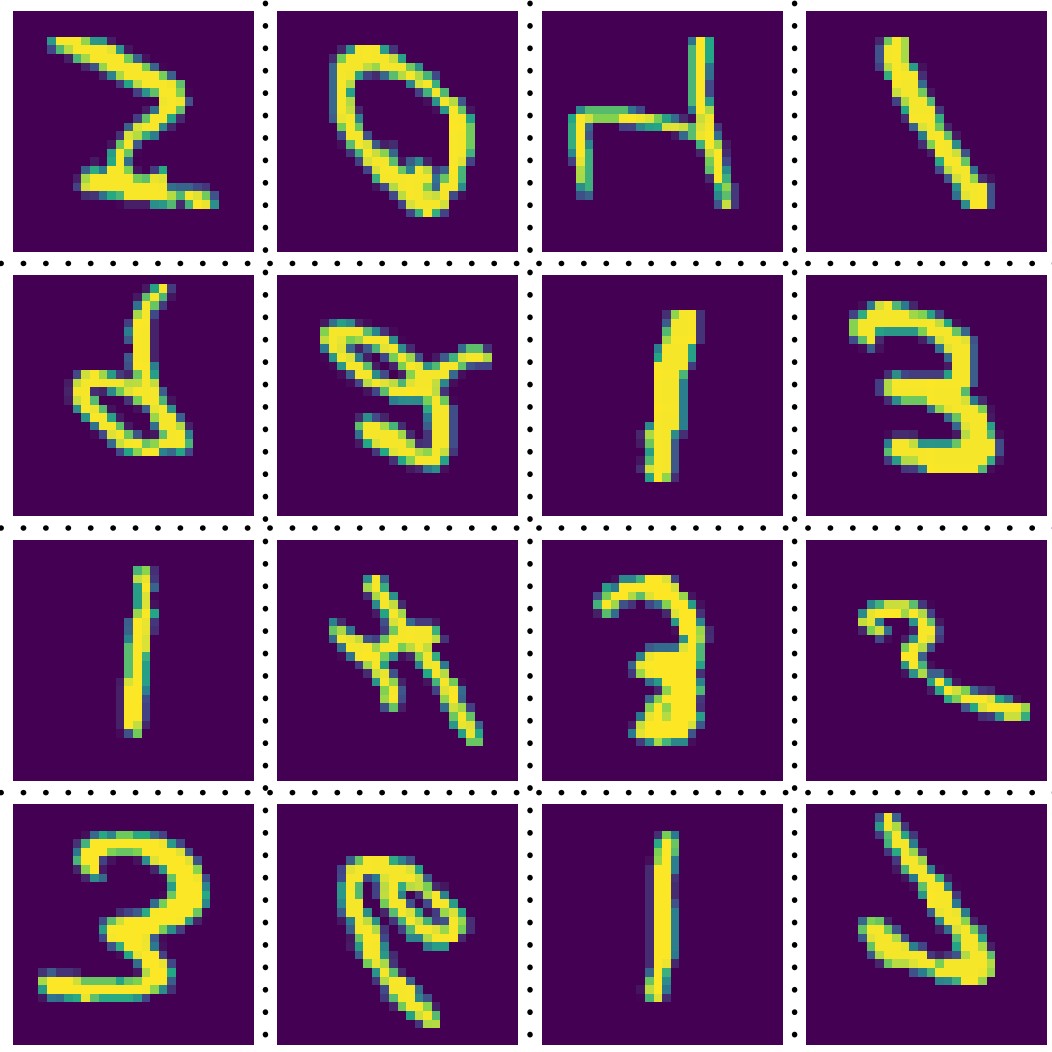

Figure 4: Examples of MNIST images LeCun et al. (1998). Each image has dimensions of $d = 28 \times 28$. The labels for the images, ordered from left to right and top to bottom, are 5, 0, 4, 1, 9, 2, 1, 3, 1, 4, 3, 5, 3, 6, 1, and 7.

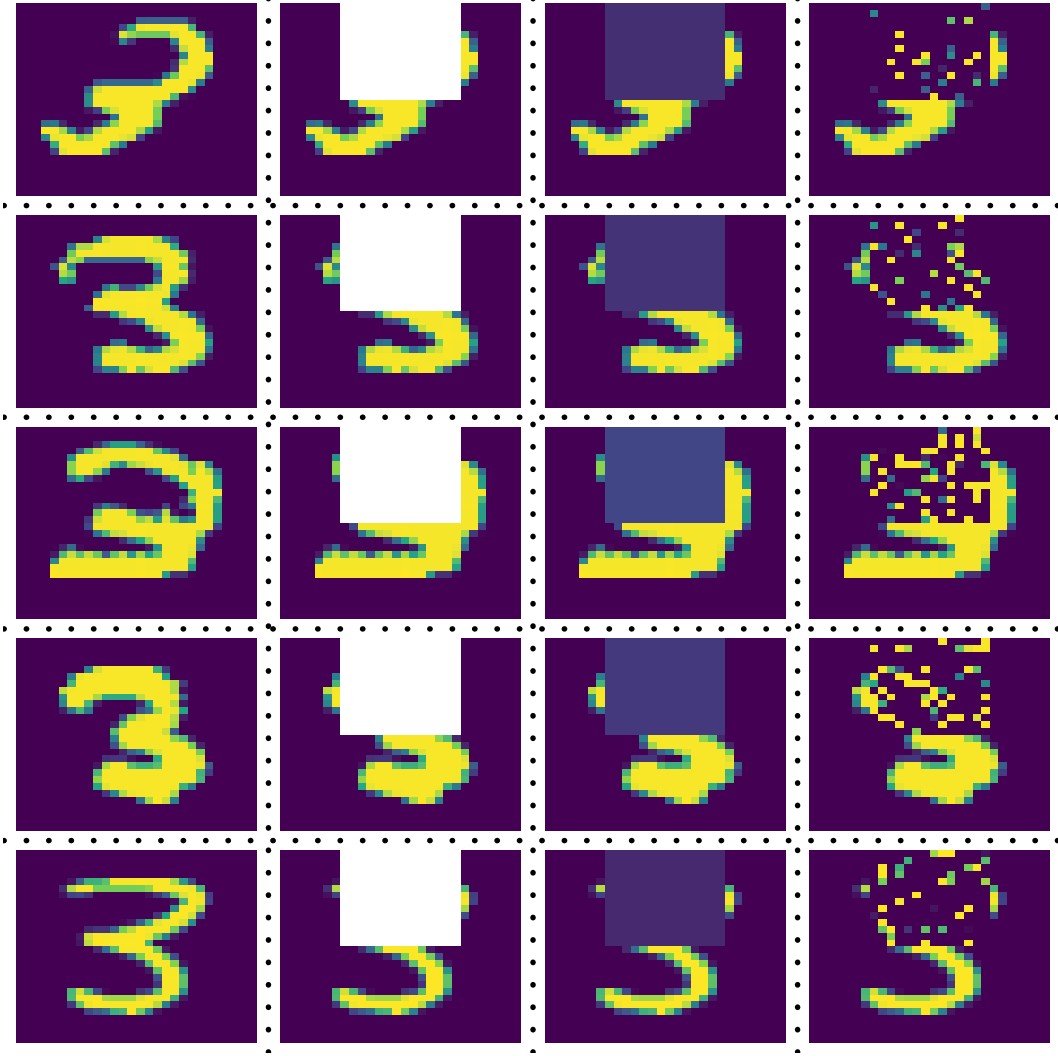

Figure 5: Examples of incompletely observed MNIST images, where white pixels represent unobserved values. Each image has dimensions of $d = 28 \times 28$ and is labelled as 3. The first column shows fully observed images; the second column shows images with upper half pixels missing (rows 1 to 14 and columns 8 to 21); the third column shows images after mean imputation; and the fourth column shows images after hot deck imputation. Each row presents different variations of the same original image.

# C  Additional Experiments

In this section, additional experiments are provided for investigating the power of proposed method. Different sample sizes and alternatives are considered compared to those in Section 5.

## C.1  Univariate Data

The missingness mechanism for producing the following two figures is the same as the mechanisms used to produce Figure 2.

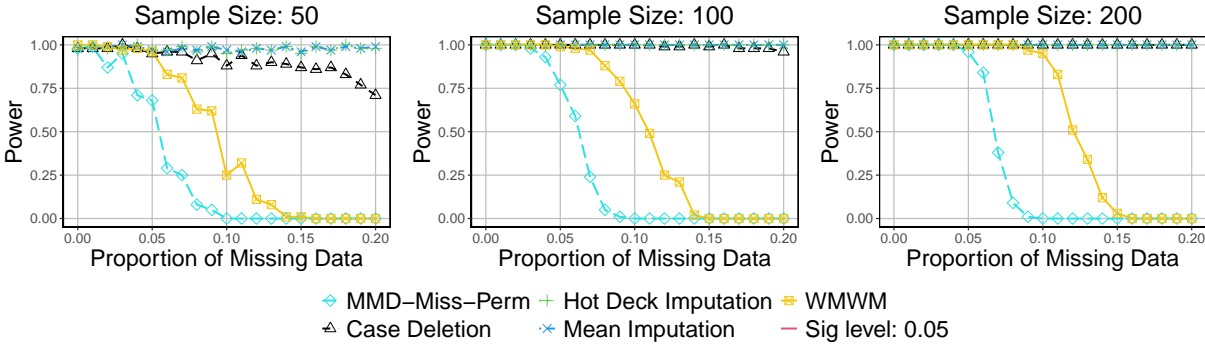

Figure 6: The power of MMD-Miss and the three common missing data approaches for univariate samples when data are missing not at random (MNAR). (Left): Sample size $n_1 = n_2 = 50$; (Middle): Sample size $n_1 = n_2 = 100$; (Right): Sample size $n_1 = n_2 = 200$. For all figures, significance level $\alpha = 0.05$, and alternative hypothesis $N(0,1)$ vs $N(1,1)$ are used. The displayed values are the average times of rejecting the null hypothesis over 100 repetitions.

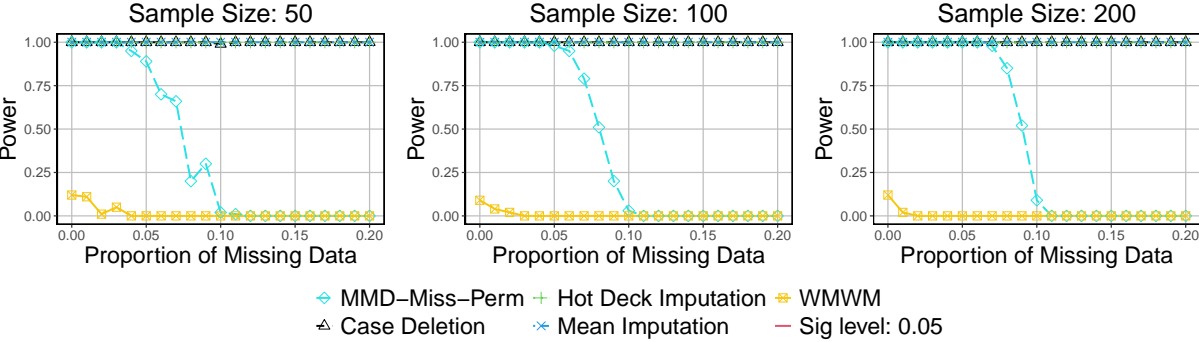

Figure 7: The power of MMD-Miss and the three common missing data approaches for univariate samples when data are missing not at random (MNAR). (Left): Sample size $n_1 = n_2 = 50$; (Middle): Sample size $n_1 = n_2 = 100$; (Right): Sample size $n_1 = n_2 = 200$. For all figures, significance level $\alpha = 0.05$, and alternative hypothesis $N(0,1)$ vs $N(0,3)$ are used. The displayed values are the average times of rejecting the null hypothesis over 100 repetitions.

## C.2 Multivariate Data

The missingness mechanism for producing the following two figures is the same as the mechanisms used to produce Figure 3.

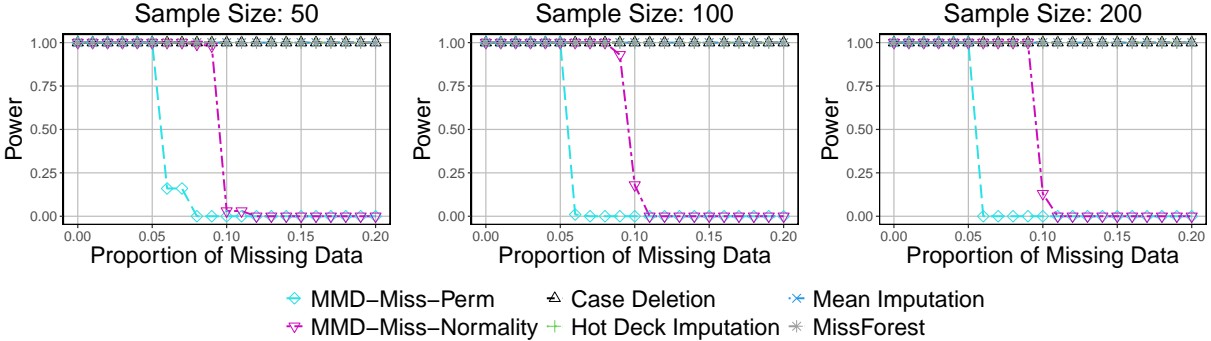

Figure 8: The power of MMD-Miss and the three common missing data approaches for multivariate samples ($d = 50$) when data are missing not at random (MNAR). (Left): Sample size $n_1 = n_2 = 50$; (Middle): Sample size $n_1 = n_2 = 100$; (Right): Sample size $n_1 = n_2 = 200$. For all figures, significance level $\alpha = 0.05$, and alternative hypothesis $N((0, \dots, 0)^T, I_{50})$ vs $N((1, \dots, 1)^T, I_{50})$ are used. The displayed values are the average times of rejecting the null hypothesis over 100 repetitions.

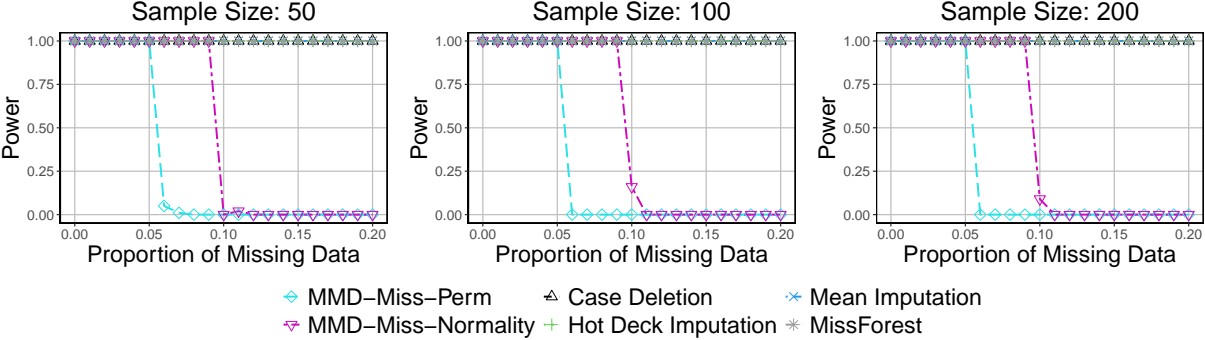

Figure 9: The power of MMD-Miss and the three common missing data approaches for multivariate samples ($d = 50$) when data are missing not at random (MNAR). (Left): Sample size $n_1 = n_2 = 50$; (Middle): Sample size $n_1 = n_2 = 100$; (Right): Sample size $n_1 = n_2 = 200$. For all figures, significance level $\alpha = 0.05$, and alternative hypothesis $N((0, \dots, 0)^T, I_{50})$ vs $N((0, \dots, 0)^T, 9I_{50})$ are used. The displayed values are the average times of rejecting the null hypothesis over 100 repetitions.

