# OpenReview forum: "MMD Two-sample Testing in the Presence of Arbitrarily Missing Data"
_TMLR — Accepted by TMLR_

### Review · Reviewer_uzLL · 2025-08-06

**Summary Of Contributions:**

The authors have introduced MMD-Miss, an extension of the Maximum Mean Discrepancy (MMD) two-sample test to handle datasets with arbitrarily missing values, without imposing assumptions on the missingness mechanisms and by leveraging the Laplacian kernel. They derive lower and upper bounds of the MMD statistic with missing values present for univariate and multivariate data. They use these bounds to control for Type I error under arbitrary missingness, employing upper bounds of p-values computed using permutations and with normality approximations. Their simulations on synthetic Gaussian data and MNIST images demonstrate that MMD-Miss controls Type I error under missing data compared to baselines and it maintains good power for 5-10% of missingness.

**Audience:**

Yes

**Claims And Evidence:**

Yes

**Requested Changes:**

See above

**Strengths And Weaknesses:**

Strengths
- Unlike prior work which is only univariate and detects only location shifts, this method supports multivariate data and identifies any distributional differences
- The work has theoretical rigour, with precise bounds on the empirical MMD statistic derived and claims justified theoretically
- The assumption-free approach is valuable for real-world applications when missingness mechanisms are unknown, and the conservatism could ensure reliability in sensitive domains


Weaknesses
- The authors have relied solely on the Laplacian kernel. Would it be possible to justify the selection and discuss this in the context of other kernels?
- The simulations could be strengthened with real-world datasets with natural missingness like clinical or survey data
- Minor typos like section 2  Shutoh et al. (2010) proposed  _an_ method

Question to the authors: Is it possible to derive or cite how the bounds ensure Neyman Pearson optimality in the context of Neyman Pearson classification, and/or to evaluate on Neyman Pearson classification benchmarks e.g. loan default or disease diagnosis datasets with induced or natural missingness to further demonstrate practical utility?

---

> ### Author Response · Authors · 2025-09-19
> **Response to Reviewer uzLL**
>
> Thank you for being positive about our work and and providing helpful feedback. We have attempted to address all suggestions as outlined below.
>
> > The authors have relied solely on the Laplacian kernel. Would it be possible to justify the selection and discuss this in the context of other kernels?
>
> We thank the reviewer for the suggestion. We now add the following paragraph in the discussion of the Maximum Mean Discrepancy in the Background section for further illustrating our choice of the Laplacian kernel:
>
> *The Laplacian and Gaussian kernels are popular choices for the MMD (Gao & Shao, 2023; Sriperumbudur etal., 2011), mainly because they are both characteristic kernels (Sriperumbudur et al., 2010; 2011). There are other characteristic kernels, such as the Matérn family of kernels (Sriperumbudur et al., 2010). In the univariate case, the MMD test has been shown to perform similarly for the Laplacian and Gaussian kernels (Bodenham & Kawahara, 2023).*
>
> *In this work we only use the Laplacian kernel $k_L$ because it allows us to derive appropriate bounds for our method. Using a different kernel with this approach may be possible but would require the derivation of different bounds. While this may seem restrictive, we emphasise that the Laplacian kernel is characteristic and so can be used with the MMD to detect any change in distribution.*
>
> > The simulations could be strengthened with real-world datasets with natural missingness like clinical or survey data.
>
> We thank the reviewer for the suggestion. We understand the importance of a method being effective on real-world datasets, but we decided to focus on showing the performance on simulated data, since it is then better to understand how the power is related to the difference (e.g. Kullback-Leibler divergence) between the two samples. We did run experiments on the MNIST data, which is a real-world data set which has been used as a benchmark in other papers on MMD (e.g. Liu et al., ICML2020). While the data may not be naturally missing in this case, we are able to control the missingness and better evaluate and analyse the performance of the method.
>
> > Minor typos like section 2 Shutoh et al. (2010) proposed an method.
>
> Thank you for pointing this out. We have carefully reviewed the manuscript and tried to remove any typos and grammatical issues.
>
> > Question to the authors: Is it possible to derive or cite how the bounds ensure Neyman Pearson optimality in the context of Neyman Pearson classification, and/or to evaluate on Neyman Pearson classification benchmarks e.g. loan default or disease diagnosis datasets with induced or natural missingness to further demonstrate practical utility?
>
> Thank you for your suggestion. The MMD test is essentially a distance measure and is used for hypothesis testing. It is unclear how to use to use it for classification. We reviewed recent work which considers Neyman Pearson classification (Tong et al., 2020), and although this work is interesting and shares concepts with hypothesis testing (Type I and Type II errors), it is fundamentally different and requires labelled data, while the MMD does not use labelled data.
>
> **References**
>
>  - Gao, H., and Shao, X. (2023) "Two sample testing in high dimension via maximum mean discrepancy." Journal of Machine Learning Research 24.304: 1-33.
>  - Bodenham, D. A., and Kawahara, Y. (2023) "euMMD: efficiently computing the MMD two-sample test statistic for univariate data." Statistics and Computing 33.5: 110.
>  - Sriperumbudur, B. K., Fukumizu, K., and Lanckriet, G. R. G. (2011) "Universality, Characteristic Kernels and RKHS Embedding of Measures." Journal of Machine Learning Research 12.7.
>  - Sriperumbudur, B. K., Gretton, A., Fukumizu, K., Schölkopf, B., & Lanckriet, G. R. (2010). Hilbert space embeddings and metrics on probability measures. The Journal of Machine Learning Research, 11, 1517-1561.
>  - Liu, F.,  Xu, W.,  Lu, J., Zhang, G., Gretton, A. and Sutherland, D. J. (2020) Learning Deep Kernels for Non-Parametric Two-Sample Tests. Proceedings of the 37th International Conference on Machine Learning, PMLR, 2020.
>  - Tong, X., Xia, L., Wang, J. and Feng, Y. (2020) Neyman-Pearson classification: parametrics and sample size requirement. Journal of Machine Learning Research, 21(12), pp.1-48.

---

### Review · Reviewer_u2qJ · 2025-08-06

**Summary Of Contributions:**

This paper proposes MMD-Miss, a novel two-sample testing method based on the Maximum Mean Discrepancy (MMD) that is designed to handle missing data in both univariate and multivariate settings, without making assumptions on the missingness mechanism (e.g., MCAR, MAR, MNAR). Unlike traditional approaches that rely on data imputation or deletion, MMD-Miss computes mathematically rigorous bounds on the MMD test statistic by accounting for all possible values of missing data. It extends recent work that derived bounds for the Wilcoxon-Mann-Whitney test in the univariate setting, enabling broader application to any distributional shift and multivariate data.
The proposed method:
* Uses the Laplacian kernel to ensure sensitivity to general distributional differences;
* Provides a permutation-based test or asymptotic approximation (via Gao & Shao, 2023) to compute p-values;
* Is provably valid in terms of Type I error control, even when the data are missing not at random (MNAR);
* Demonstrates strong statistical power (particularly for 5–10% missing data) through numerical simulations;
* Highlights the failure of standard imputation or deletion techniques to control Type I error under MNAR scenarios, even with low missingness.
To the best of the authors' knowledge, MMD-Miss is the first MMD-based two-sample test with formal Type I error guarantees under arbitrary missingness, including in high-dimensional multivariate data.

**Audience:**

Yes

**Claims And Evidence:**

Yes

**Requested Changes:**

- Introduction: "*Nearly all two-sample testing methods are designed solely for data that are fully observed.*": Which tests are applicable to data with missing values? Have such methods been included in the benchmark? This statement should be substantiated with more detail or appropriate citations.
- Introduction: paragraph "two-sample testing ...": This paragraph in question appears somewhat misplaced. Typically, introductions conclude with a summary of the main contributions and, ideally, an outline of the paper’s structure. Consider moving this paragraph to a more appropriate section (e.g., related work or discussion), or rephrasing it to fit more naturally at the end of the introduction if it is meant to serve as motivation.
- Related work: While the paper is well-situated within the existing literature on two-sample testing with missing data, a few relevant and recent works seem to be missing from the discussion. In particular, "Two-sample Testing Using Deep Learning" [1] and "A Permutation-Free Kernel Two-Sample Test" [2] may offer complementary perspectives or contrasting approaches that would strengthen the related work section. Including these references would help position the contribution more clearly with respect to the state of the art.

[1] Kirchler, M., Khorasani, S., Kloft, M., & Lippert, C. (2020, June). Two-sample testing using deep learning. In International Conference on Artificial Intelligence and Statistics (pp. 1387-1398). PMLR.

[2] Shekhar, S., Kim, I., & Ramdas, A. (2022). A permutation-free kernel two-sample test. Advances in Neural Information Processing Systems, 35, 18168-18180.
- Background: The current structure of the paper presents subsections somewhat in isolation, without sufficient transitions or narrative flow to guide the reader. As this is a scientific article rather than a technical note, it is important to articulate clearer motivations and provide smoother transitions between the different components of the method and experiments. For example, it would be beneficial to briefly explain the purpose of each section and how it builds upon the previous one, making the contributions and rationale behind the methodological choices more accessible and coherent. Reorganizing or enriching the structure with guiding commentary would significantly improve readability and clarity.
- Maximum Mean Discrepancy:The Maximum Mean Discrepancy (MMD) test is introduced in a technical manner, but the paper would benefit from a more accessible presentation of its core principle, strengths, and limitations, particularly in comparison to classical alternatives such as the Wilcoxon-Mann-Whitney or Kolmogorov-Smirnov tests.
( MMD² formula : "n1" for the second sum
- "al., 2024). Two popular choices of characteristic kernels are Laplacian and Gaussian kernels" : could you justify this point (with references if possible) ?
- "Our method is based on the MMD with the Laplacian kernel" : why? this choice must be more justified (beyond technical reasons for demonstrations)
- Permutation Test: Please explain the principles of the test and its advantages (non-parametric distribution, etc.).
- Bounding MMD with Missing Data: The section on bounding the MMD statistic in the presence of missing data is technically sound, but would benefit from a clearer explanation of the rationale behind this approach. Specifically, it is important to guide the reader by explicitly stating why bounding is necessary
- Lemma1 : "n1" for the second sum
- Multivariate data : Is it possible to have different dimensions for the variables? If so, this is a limitation of the test, which must be indicated.
- Bounding p-value using normality approximation: In which cases is it preferable to use this approximation?
- Experiments :
    - the XP protocols are not very clear and difficult to follow.
    - Would it be possible to add sensitivities with a missing data rate varying from 0% to 5%? For % rates, it would be interesting to compare the proposed approach with conventional tests.
    - Why is the method based on the Wilcoxon-Mann-Whitney test (work by Zeng et al.) not added as a competitor in the univariate case?
    - From my understanding, only independent variables are simulated, in order to test type I error. Would it be possible to play scenarios with dependent variables (in a complex way) to analyze the type II error? d'autant plus qu'il est dit en background "a two-sample testing method is preferred if it has a lower probability of making a Type II error"
    - The imputation methods are not very efficient for evaluating the method. Would it be possible to add more efficient methods (missForest, for example)?
    - MNIST dataset: "To assess Type I error, datasets X and Y are generated by randomly sampling with replacement from MNIST training set images labeled as 3." Why 3 and not another? Is it possible to carry out the experiment with other labels?
- Conclusion: The conclusion is far too short. It should include limitations (e.g. choice of Laplace kernel, assumptions in the demonstrations, etc.) as well as perspectives (possibility of extending this approach to the Gaussian kernel? data in large dimensions, etc.).

**Strengths And Weaknesses:**

**Strengths**
- Good paper structure
- Deals with an interesting and practical subject
- clear objective and positioning of the paper in relation to the state of the art
- easy-to-follow mathematical demonstration
- Satisfactory performance

**Weaknesses**
- Some methodological choices are not properly justified (choice of kernel, etc.).
- Principles, advantages and disadvantages need to be highlighted, particularly in relation to literature.
- Contributions not highlighted
- experiments are difficult to follow
- The paper could be improved in terms of clarity and transitions between sections, in order to better guide the reader through the narrative.
- A few clarifications and modifications to be made (see requested changes section)

---

> ### Author Response · Authors · 2025-09-19
> **Response to Reviewer u2qJ**
>
> Thank you for your constructive feedback. We have addressed all comments and requested changes as outlined below.
>
> > Introduction: *"Nearly all two-sample testing methods...*
>
> Thank you for the suggestion. The methods discussed in Related Work in Section 2 (Background) are applicable to data with missing values, although they all make missing data assumptions, such as missing completely at random or missing at random. We now modify the sentence in the introduction to follows:
>
> *Nearly all two-sample testing methods are designed solely for data that are fully observed, with a few exceptions discussed in the next section.*
>
> We evaluate the performance of MMD after using different imputation methods and compare to Zeng et al. (2024) which does not make missing data assumptions, but is only applicable for detecting a change in location for univariate data. Other approaches make assumptions about the missingness mechanism or the values of the missing data.
>
>  > Introduction: paragraph "two-sample testing ..."...
>
> Thank you for the suggestion. We now rephrase the final paragraph of the Introduction to better summarise the motivation and the contributions of our work:
>
> *Our work is motivated to ensure...without making any assumptions about the missing data.*
>
> > Related work: While the paper...
>
> Thank you for these references. We now include the following paragraph in Related Work in Background section:
>
> *Various authors have proposed extensions...about the underlying missingness mechanism.*
>
> > Background: The current structure of the paper...
>
> Thank you for your comment. We have reorganized the structure of the Background section to improve its narrative flow and overall clarity. To better guide the reader through the content, we now begin the section with the following introductory paragraph:
>
> *This section provides the necessary background for our work. We begin by...as well as extensions of the MMD test.*
>
> > Maximum Mean Discrepancy: The Maximum Mean Discrepancy (MMD) test is introduced...
>
> Thank you for the suggestion. We now introduce the MMD test by first starting with the following sentence for a more accessible presentation:
>
> *The Maximum Mean Discrepancy (MMD) is a two-sample test used for deciding if two groups of independent observations $X=\{x_1,…,x_{n_1}\}$ and $Y=\{y_1,…,y_{n_2}\}$ are sampled from different distributions $p$ and $q$.*
>
> Furthermore, we now add the following paragraph “Strengths and limitations of the MMD” to better explain the core principle, strengths, and limitations of the MMD test and compare it with classical alternatives such as the Wilcoxon-Mann-Whitney or Kolmogorov-Smirnov tests:
>
> *As discussed, the core principle of MMD is...which has been shown to generally perform well.*
>
> > Two popular choices of characteristic kernels are Laplacian and Gaussian kernels" could you justify this point (with references if possible)?
>
> We thank the reviewer for the suggestion. This point that Laplacian and Gaussian kernels are popular are used by several different authors directly, including Sriperumbudur et al. (2011), Gao & Shao (2023). We now justify this point by adding the following paragraph:
>
> *The Laplacian and Gaussian kernels are popular choices for the MMD (Gao & Shao, 2023; Sriperumbudur etal., 2011), mainly because they are both characteristic kernels (Sriperumbudur et al., 2010; 2011). There are other characteristic kernels, such as the Matérn family of kernels (Sriperumbudur et al., 2010). In the univariate case, the MMD test has been shown to perform similarly for the Laplacian and Gaussian kernels (Bodenham & Kawahara, 2023).*
>
> *In this work we only use the Laplacian kernel $k_L$ because it allows us to derive appropriate bounds for our method. Using a different kernel with this approach may be possible but would require the derivation of different bounds. While this may seem restrictive, we emphasise that the Laplacian kernel is characteristic and so can be used with the MMD to detect any change in distribution.*
>
> > "Our method is based on the MMD with the Laplacian kernel..."...
>
> Please see comment above.
>
> **References**
>  - Zeng, Y.,  Adams, N. M. and Bodenham, D. A. (2024) On two-sample testing for data with arbitrarily missing values, arXiv preprint arXiv:2403.15327
>  - Gao, H., and Shao, X. (2023) "Two sample testing in high dimension via maximum mean discrepancy."  - - Journal of Machine Learning Research 24.304: 1-33.
>  - Bodenham, D. A., and Kawahara, Y. (2023) "euMMD: efficiently computing the MMD two-sample test statistic for univariate data." Statistics and Computing 33.5: 110.
>  - Sriperumbudur, B. K., Fukumizu, K., and Lanckriet, G. R. G. (2011) "Universality, Characteristic Kernels and RKHS Embedding of Measures." Journal of Machine Learning Research 12.7.
>  - Sriperumbudur, B. K., Gretton, A., Fukumizu, K., Schölkopf, B., & Lanckriet, G. R. (2010). Hilbert space embeddings and metrics on probability measures. The Journal of Machine Learning Research, 11, 1517-1561.

---

> > ### Author Response · Authors · 2025-09-19
> > **Response to Reviewer u2qJ (continued)**
> >
> > > Permutation Test: Please explain the principles...
> >
> > Thank you for your comment. We now start the discussion of the Permutation Test section with the following paragraph:
> >
> > *The permutation test is a common numerical procedure...only asymptotic true.*
> >
> > > Bounding MMD with Missing Data:...
> >
> > Thank you for the suggestion. We have modified the first paragraph of Section 3 to better explaining why bounding is necessary:
> >
> > *In this section, we provide bounds...to detect any change in distribution.*
> >
> > > Lemma1 : "n1" for the second sum.
> >
> > Thank you for spotting this typo. We have now corrected it in the revised version of the manuscript.
> >
> > > Multivariate data : Is it possible to have different dimensions...
> >
> > In two-sample testing, it is ubiquitous in the literature for the data $X$ and $Y$ to have the same dimension $d$. Therefore, we focused only on this case. Moreover, the MMD test, for computational reasons, requires the two samples to have the same dimension, and we now emphasise this in the Multivariate Data section:
> >
> > *Note that $X$ and $Y$ are assumed to have the same dimension $d$.*
> >
> > > Bounding p-value using normality approximation:...
> >
> > Thank you for the suggestion. As we discussed in the Normal Approximation section in the Background, the normal approximation is only suitable for sample sizes $n, m \geq 25$ with dimension $ \geq 50$ (Gas and Shao, 2023). Our experiments results show that when these conditions are satisfied, using this approximation (MMD-Normality) increases the statistical power compared to the proposed method based on the permutation approach (MMD-Perm). We now emphasize this point in the Conclusion section of the main text:
> >
> > Our experiments show that in high-dimensional situations, i.e. $d \geq 50$, using a normal approximation (Gao and Shao, 2023) (MMD-Normality) improves the statistical power of MMD-Miss based on permutation test (MMD-Perm).
> >
> > > Experiments: the XP protocols...
> >
> > We thank the reviewer for this comment. We have now rewritten the description of the experiments in Section 5 to better highlight the goal of each experiment.
> >
> > > Experiments: Would it be possible to add sensitivities...
> >
> > Thank you for this suggestion. Figure 2 shows the Type I error and statistical power of the various methods as the missing data increases from 0% to 20%. One of the purposes of these experiments is to show that using imputation methods with the MMD will result in the Type I error not being controlled. In Zeng et al. (2024) it is shown that when the Wilcoxon-Mann-Whitney (WMW) test is used with various imputation methods, the Type I error will fail to be controlled. It is likely that this would also occur if the WMW were replaced with another test, such as Student’s t-test, since ignoring or imputing the missing data is what causes the test’s Type I error not to be controlled, rather than the mechanism of the test itself.
> >
> > > Experiments: Why is the method based on the Wilcoxon-Mann-Whitney test (work by Zeng et al.)...
> >
> > Thank you for this suggestion. We have now added the WMWM method (Zeng et al.) to the experiments with results displayed in Figures 1 and 2.
> >
> > > Experiments: From my understanding, only independent variables are simulated...
> >
> > Thank you for this comment. Most two-sample tests, including the t-test, WMW and MMD tests, consider the observations to be independent. If this assumption were not valid, it would be theoretically difficult to derive a procedure which is guaranteed to control the Type I error.
> >
> > > Experiments: The imputation methods are not very efficient for evaluating the method...more efficient methods (missForest, for example)?
> >
> > Thank you for this suggestion. We have added MissForest (Stekhoven & Bühlmann, 2012) to the results in Figures 1 and 3. Note that MissForest is only suitable for multivariate data. Our experiments show that, along with other imputation methods, using MissForest will result in the Type I error not being controlled.
> >
> > > Experiments: MNIST dataset:...
> >
> > Thank you for this comment. From the MNIST dataset, we compared samples of 0s and 3s. We could have chosen another pair of digits, but our goal was simply to illustrate that the method could be applied to such real-world image data. It is possible to do experiments on different pairs of labels, but it would not add anything significant to the results.
> >
> > > Conclusion: The conclusion is far too short...
> >
> > Thank you for this suggestion. We have rewritten and expanded the conclusion to include a discussion of limitations and perspectives as suggested.
> >
> > **References**
> >
> >  - Gao, H., Xiaofeng Shao, X. "Two sample testing in high dimension via maximum mean discrepancy." Journal of Machine Learning Research 24.304 (2023): 1-33.
> >  - Zeng, Y., Adams, N. M. and Bodenham, D. A. (2024) On two-sample testing for data with arbitrarily missing values, arXiv preprint arXiv:2403.15327
> >  - Stekhoven, D.J., Bühlmann, P., 2012. MissForest—non-parametric missing value imputation for mixed-type data. Bioinformatics, 28(1), pp.112-118.

---

> > > ### Comment · Reviewer_u2qJ · 2025-10-02
> > >
> > > I am satisfied with the authors' response and the changes they have made to address my concerns. I acknowledge their efforts to improve their paper, and this confirms my initial assessment.

---

### Review · Reviewer_EBYb · 2025-09-12

**Summary Of Contributions:**

This paper considers two-sample test by MMD (Maximum Mean Discrepancy), in which the dataset contains missing values. The basic idea is to derive the bound of MMD values for any miss values, by which the bound of the p-value can also be derived. The proposed method can provides hypothesis test that can appropriately control the Type I error without depending on the mechanism of the missing value generation (such as well-known MCAR, MAR, and MNAR). Two approaches to the null distribution calculation are provided based on the permutation and asymptotic normality, respectively. The empirical evaluation on the synthetic and MNIST datasets show that the proposed method can inhibit Type I error less than significance level.

**Audience:**

Yes

**Claims And Evidence:**

Yes

**Requested Changes:**

About the asymptotic normality, since (8) is seemingly non-negative, it is intuitively a bit weird that this quantity converges to standard normal distribution N(0,1). For example, the value of p = 1 - \Psi(MMD^2 / \sqrt{c V}) becomes p = 0.5 when MMD^2 = 0, but this seemingly indicates that p cannot becomes larger than 0.5 because MMD^2 >= 0. However, usually, p-value should be the uniform distribution in [0,1]. How is this inconsistency explained?

About the tightness of the bound, for example, the tightest upper bound of MMD^2_u(X,Y) should be \max_{all missing values} MMD^2_u(X,Y). Although I guess it is computationally difficult, could you provide some discussion about it or a relation between the tightest bound and the current bound?

In page 23, the authors derived dT(z)/dz. I guess the zero point of this derivative can be calculated analytically, i.e, dT(z)/dz = 0 can be solved wrt z. Then, if the extreme point (solution of dT(z)/dz = 0) exists in (z_3,z_4), it would be the minimum or maximum. Otherwise, The minimum or maximum should exist at z_3 or z_4. Is this logic correct? If so, by calculating it for all the intervals (the same logic would be applicable to any intervals if my guess is correct), can the exact maximum (or minimum) be derived?

In the case of the permutation approach, the inside of the indicator can be written as MMD^2_u(X_\sigma(i), Y_\sigma(i)) - MMD^2_u(X, Y) >= 0. Then, can directly considering the lower bound of the left hand side MMD^2_u(X_\sigma(i), Y_\sigma(i)) - MMD^2_u(X, Y) derive a tighter bound?

The authors first mentioned that the Laplacian and Gaussian kernel are standard, but the technical discussion is only for the Laplacian kernel throughout the paper. The reason why focusing only on Laplacian is not revealed. I think explicitly providing the reason would be beneficial for readers.

In Figure 1, Type I error of the proposed method is much less than 0.05. Does this indicate the conservative property of the proposed method? It would be better to explicitly discuss why it is not close to 0.05.

Minor:

- I cannot find how \beta in the Laplacian kernel is set.

- I cannot find the setting of the number of permutation B in MMD-Miss-Perm.

- In the first definition of MMD^2 in page 3, the left had side should not be squared? (or right hand side should be squared).

- In the plots of the power, the line at 0.05 is shown. What is it for? If there's no specific reason, it should be removed to avoid confusion with the Type I error.

**Strengths And Weaknesses:**

S: Overall, I think that the paper is well-written and the contents are scientifically valid, sufficiently.

S: The proposed method is independent from the mechanism of the missing value generation and I think this is a strong advantage in practice.

S: The derivation of the bound is based on simple reasonable approaches, by which a rationale behind why the proposed method works without assuming missing mechanism is clear.

W: The tightness of the bound is not clear. Although it would be tighter than a naive bound, I guess the bound seems loose. It is unclear how the bound looseness affects the decrease of the power of hypothesis test.

W: Only the Laplacian kernel can be used in the kernel function in MMD, which may be a bit restrictive (e.g., well-known Gaussian kernel is not discussed).

---

> ### Author Response · Authors · 2025-09-19
> **Response to Reviewer EBYb**
>
> Thank you for being positive about our work and providing constructive feedback. We have addressed all comments and suggestions as outlined below.
>
> > About the asymptotic normality, since (8) is seemingly non-negative, it is intuitively a bit weird that this quantity converges to standard normal distribution N(0,1). For example...
>
> We thank the reviewer for the comment. We agree that the $MMD^2$ between two distributions p and q should be non-negative, and when the null hypothesis is true, i.e. $p = q$, the $MMD^2$ between $p$ and $q$ is $0$, i.e. $MMD^2(p,q) = 0$. This is discussed in the background section. However, $MMD^2_u(X,Y)$ in Equation (8) is an unbiased estimate of $MMD^2(p,q)$, and can be negative in finite samples due to sampling noise.
>
> > About the tightness of the bound, ...
>
> We thank the reviewer for the suggestion. We now emphasize in Lemma 2 that our bounds of function $T(z)$ for $A_3$ and $A_4$ are tight. Meanwhile, we include the following Remark 1 in the main paper for discussing the tightest bound and the current bound:
>
> *As we discussed previously, our strategy for bounding the MMD test statistic $MMD^2_u(X,Y)$ is to provide the bounds for the terms $A_1$, $A_2$, $A_3$ and $A_4$, separately. Since $A_1$ includes only the unobserved data in X and Y, the bounds of it can be constructed according to Inequality (2) and cannot be improved. The second term $A_2$ includes only observed data. The third and the fourth term $A_3$, $A_4$ can be decomposed into the sum of functions of $T(x)$, as shown in Inequality (3).  We show in Lemma 2 that the tight lower and upper bounds of $T(x)$ can be constructed. Hence the bounds of term $A_3$ and $A_4$ are tight. While all the bounds of $A_1$, $A_2$, $A_3$ and $A_4$ are tight, our current bounds for the MMD test statistic are not tight since we provide the bounds of $A_1$, $A_2$, $A_3$, $A_4$ separately, rather than jointly.*
>
> > In page 23, the authors derived dT(z)/dz...
>
> We thank the reviewer for the suggestion. This logic is correct and very similar to that used in our proof. We show that $T(z_0) \ge \min$ \{$0, T(z_3), T(z_4)$ \}, hence there is no need to evaluate the analytical solution of $dT(z)/dz = 0$ wrt $z$. We now emphasize that Lemma 2 in the main text already provides tight bounds for the function $T(z)$, since $z$ can take any value in \{ $x_1, … x_{l_1}, y_1, ..., y_{l_2}$ \}, and in particular when $z$ goes to infinity, $T(z)$ goes to $0$. We now add the following sentence in the end of Lemma 2:
>
> *Moreover, these bounds for $T(z_0)$ are tight, and we have equality when$ z_0 \in$ \{$x_1, … x_{l_1}, y_1, ..., y_{l_2}$\} or as $z_0 \to \pm \infty$.*
>
> We note that while the bounds of $T(z)$ are tight, it does not directly lead to the tight bounds of the MMD test statistic, since $T(z)$ is only applied for bounding the term $A_3$ in MMD test statistic. As described in Lemma 1, the full MMD test statistic involves several addition components, i.e. $A_1$, $A_2$, $A_3$, and $A_4$. We now mention this point in Remark 1 in the main text.
>
> > In the case of the permutation approach,...
>
> We thank the reviewer for the insightful suggestion. We agree that, in the permutation approach, by considering the bounds of $MMD^2_u(X_\sigma(i), Y_\sigma(i)) - MMD^2_u(X, Y)$ directly, rather than that for $MMD^2_u(X_\sigma(i), Y_\sigma(i))$ and $MMD^2_u(X, Y)$ separately could potentially lead to a tighter bound. However, this is a substantially more challenging technical problem. Our current approach for bounding the MMD test statistic relies on first decomposing the MMD test statistics into four parts $A_1$, $A_2$, $A_3$, $A_4$ according to whether the data are observed in $X$ and $Y$. Importantly, using Lemma 2, we provide bounds of $A_3$ and $A_4$, where only values in one sample, e.g. $X$, are missing, but the other sample is completely observed. This is already challenging, but we have shown this approach to be feasible.
>
> The difficulty of bounding the difference $MMD^2_u(X_\sigma(i), Y_\sigma(i)) - MMD^2_u(X, Y)$ directly is that one value in sample X in the original data could move to sample Y after permutation (and vice versa). Hence, constructing a bound for the difference directly might require us to consider additional results which can distinguish values that change samples after permutation, e.g. from $X$ to $Y$, from values that do not, although this is technically nontrivial. We appreciate the reviewer’s insightful suggestion and leave this for future work.

---

> > ### Author Response · Authors · 2025-09-19
> > **Response to Reviewer EBYb (continued)**
> >
> > > The authors first mentioned that the Laplacian and Gaussian kernel are standard, but the technical discussion is only for the Laplacian kernel throughout the paper. The reason why focusing only on Laplacian is not revealed. I think explicitly providing the reason would be beneficial for readers.
> >
> > We thank the reviewer for the suggestion. We now add the following paragraph in the discussion of the Maximum Mean Discrepancy in the Background section to further justify our choice of the Laplacian kernel:
> >
> > *The Laplacian and Gaussian kernels are popular choices for the MMD (Gao & Shao, 2023; Sriperumbudur etal., 2011), mainly because they are both characteristic kernels (Sriperumbudur et al., 2010; 2011). There are other characteristic kernels, such as the Matérn family of kernels (Sriperumbudur et al., 2010). In the univariate case, the MMD test has been shown to perform similarly for the Laplacian and Gaussian kernels (Bodenham & Kawahara, 2023).*
> >
> > *In this work we only use the Laplacian kernel $k_L$ because it allows us to derive appropriate bounds for our method. Using a different kernel with this approach may be possible but would require the derivation of different bounds. While this may seem restrictive, we emphasise that the Laplacian kernel is characteristic and so can be used with the MMD to detect any change in distribution.*
> >
> > > In Figure 1, Type I error of the proposed method is much less than 0.05. Does this indicate the conservative property of the proposed method? It would be better to explicitly discuss why it is not close to 0.05.
> >
> > We thank the reviewer for the suggestion. We now include the following Remark 2 on page 13 of the experiments section:
> >
> > *As reflected in these results, our proposed method is guaranteed to control the Type I error, even in cases where data are missing not at random. However, the cost is that our method is conservative, since it only declares a significant  result when the maximum possible $p$-value, considered over all possible imputations, is significant. When the proportion of missing data increases, the upper and lower bounds of the MMD test statistic increase and decrease, respectively, and so the power decreases.*
> >
> > **Minor**
> > > I cannot find how \beta in the Laplacian kernel is set.
> >
> > We use the Median Heuristic method for determining the beta. This is discussed in Appendix B.2. In the first paragraph of Section 5 we now write:
> >
> > *For MMD-Miss, the parameter $\beta$ in the Laplacian kernel is chosen using the median heuristic, which generally works well (Gretton et al., 2012, Bodenham & Kawahara, 2023) and is described in Appendix B.2.*
> >
> > > I cannot find the setting of the number of permutation B in MMD-Miss-Perm.
> >
> > The number of permutations is set to $B = 100$. In the first paragraph of Section 5 we now write:
> >
> > *The number of permutations used for MMD-Perm and the imputation methods is set to $B = 100$, as described in Appendix B.2.*
> >
> > > In the first definition of MMD^2 in page 3, the left had side should not be squared? (or right hand side should be squared).
> >
> > We thank the reviewer for spotting this typo. The right hand side of the equation has now been squared.
> >
> > > In the plots of the power, the line at 0.05 is shown. What is it for? If there's no specific reason, it should be removed to avoid confusion with the Type I error.
> >
> > We thank the reviewer for the suggestion. The line at $0.05$ power has been removed for avoiding confusion with the Type I error.
> >
> > **References**
> >
> >  - Gao, H., and Shao, X. (2023) "Two sample testing in high dimension via maximum mean discrepancy." Journal of Machine Learning Research 24.304: 1-33.
> >  - Sriperumbudur, B. K., Fukumizu, K., and Lanckriet, G. R. G. (2011) "Universality, Characteristic Kernels and RKHS Embedding of Measures." Journal of Machine Learning Research 12.7.
> >  - Sriperumbudur, B. K., Gretton, A., Fukumizu, K., Schölkopf, B., & Lanckriet, G. R. (2010). Hilbert space embeddings and metrics on probability measures. The Journal of Machine Learning Research, 11, 1517-1561.
> > - Bodenham, D. A., and Kawahara, Y. (2023) "euMMD: efficiently computing the MMD two-sample test statistic for univariate data." Statistics and Computing 33.5: 110.
> >  - Gretton, A., Borgwardt, K.M., Rasch, M.J., Schölkopf, B. and Smola, A., 2012. A kernel two-sample test. The journal of machine learning research, 13(1), pp.723-773.

---

### Decision · Action_Editor_idEB · 2025-11-20

**Recommendation:** Accept as is

**Additional Comments:**

Congratulations to the authors on a fine piece of work!

**Audience:**

Yes

**Audience Explanation:**

There is a consensus among reviewers this is the case.

**Claims And Evidence:**

Yes

**Claims Explanation:**

There is a consensus among reviewers this is the case.